# ARIA: Training Language Agents with Intention-Driven Reward Aggregation

**Ruihan Yang**[1,♡]*, **Yikai Zhang**[2,♡]*†, **Aili Chen**[2,♡], **Xintao Wang**[2,♡],
**Siyu Yuan**[1,♡], **Jiangjie Chen**[♠], **Deqing Yang**[1,♡]†, **Yanghua Xiao**[2,♡]

[1,♡]School of Data Science,Fudan University, Fudan University    [♠]ByteDance Seed
[2,♡]College of Computer Science and Artifcial Intelligence, Fudan University
[♡]{rhyang17,alchen20,yangdeqing,shawyh}@fudan.edu.cn
[♡]{syyuan21,ykzhang22}@m.fudan.edu.cn    [♠]{jiangjiec}@bytedance.com

Project Page: https://aria-agent.github.io

## Abstract

Large language models (LLMs) have enabled agents to perform complex reasoning and decision-making through free-form language interactions. However, in open-ended language action environments (*e.g.*, negotiation or question-asking games), the action space can be formulated as a joint distribution over tokens, resulting in an exponentially large action space. Sampling actions in such a space can lead to extreme reward sparsity, which brings large reward variance, hindering effective reinforcement learning (RL). To address this, we propose ARIA, a method that **A**ggregates **R**ewards in **I**ntention space to enable efficient and effective language **A**gents training. ARIA aims to project natural language actions from the high-dimensional joint token distribution space into a low-dimensional intention space, where semantically similar actions are clustered and assigned shared rewards. This intention-aware reward aggregation reduces reward variance by densifying reward signals, fostering better policy optimization. Extensive experiments demonstrate that ARIA not only significantly reduces policy gradient variance, but also delivers substantial performance gains of an average of 9.95% across four downstream tasks, consistently outperforming offline and online RL baselines.

## 1 Introduction

Large language models (LLMs) have demonstrated strong capabilities in text comprehension and generation, enabling the development of autonomous agents that operate through natural language, commonly referred to as language agents [1; 2; 3]. Language agents are increasingly expected to interact with environments through language-driven actions to accomplish diverse tasks, such as web navigation [4; 5], text-based games [6; 7; 8], and negotiation [9; 10]. These tasks often require long-horizon planning and reasoning to achieve high-level goals, posing significant challenges for current language agents [11; 12; 13; 14; 15]. According to the structure of the action space, language agent tasks can be broadly categorized into *constrained action space tasks* and *open-ended language action tasks*. The former requires agents to perform actions from a predefined, discrete, and verifiable action set, where language serves as a template or command interface to structured environments [16; 17]. In contrast, the action space of *open-ended language action tasks* comprises free-form natural language utterances without strict validity constraints [18; 19]. These tasks introduce unique challenges: *1)* Agents must generate diverse, context-sensitive language actions that dynamically influence

---

*Equal Contribution.
†Corresponding authors.

other agents or the environment. *2)* The open-endedness of language actions gives rise to a vast, unstructured, and highly strategic action space, requiring agents to reason, adapt, and optimize beyond fixed patterns. Given these challenges, we pose the following research question: ***How can we enhance the performance of language agents in open-ended language action tasks?***

Reinforcement learning (RL) is widely used to enhance language agents in complex tasks by enabling them to learn through interaction and feedback [20; 21]. However, in open-ended language action settings, RL faces serious challenges due to **extremely sparse rewards** caused by **exponentially large action space**, where actions are represented as token sequences. Given a vocabulary of size $\mathcal{V}$ and an average sequence length $\mathcal{L}$, the action space scales as $\mathcal{V}^{\mathcal{L}}$, resulting in a combinatorial and exponential explosion. Existing methods directly assign environmental rewards by averaging or decaying. Yet these are inadequate for open-ended tasks, where sampling-based methods such as PPO [22] and REINFORCE [23] must search a vast, unstructured space under sparse and delayed rewards. This leads to **high variance in reward estimation** and **inefficient policy optimization**.

To address these challenges, we propose **semantic projection**, which projects actions from the high-dimensional token space into a low-dimensional intention space, enabling reward aggregation across semantically equivalent actions. LLM agents' actions often reflect underlying intentions, which are far fewer than token combinations. For example, the utterances "*I will concede first in order to encourage my opponent to compromise*" and "*By taking the initiative to compromise, I aim to prompt my counterpart to do the same.*" convey the same intention of prompting compromise through concession. By grouping such actions under shared intentions, we reduce the action space from $\mathcal{V}^{\mathcal{L}}$ to intention space $\mathcal{C}$, where $|\mathcal{C}| \ll |\mathcal{V}^{\mathcal{L}}|$. This transformation reduces variance by densifying sparse rewards, and facilitates more efficient policy optimization.

Building on semantic projection, we propose ARIA, a method that **A**ggregates **R**ewards in **I**ntention space for efficient training of language **A**gents. ARIA maps natural language actions into a task-specific intention space via **semantic projection**, enabling reward aggregation across semantically similar actions to stabilize and improve policy learning. To automatically construct the intention space $\mathcal{C}$, ARIA applies hierarchical clustering [24] over sentence embeddings and adaptively adjusts the clustering granularity. It then aggregates rewards for actions sharing similar intentions and uses REINFORCE [23] to optimize the policy over this compressed space. We evaluate ARIA on four language action tasks, including two single-agent games (*Guess My City*, *20 Questions*) and two adversarial games (*Negotiation*, *Bargaining*). Experimental results show that: *1)* ARIA significantly reduces reward variance, enabling stable training and improved policy gradient efficiency; *2)* It consistently outperforms offline and online RL baselines, achieving an average improvement of 9.95% across all tasks.

In summary, our key contributions are as follows: *1)* We propose the operation of **semantic projection**, which projects actions from the high-dimensional token sequence space into a compact intention space, effectively mitigating reward sparsity in free-form language action tasks; *2)* Built upon **semantic projection**, we design ARIA, a principled approach for training language agents with intention-driven reward aggregation; *3)* We conduct extensive experiments on both single-agent and adversarial tasks, showing that ARIA reduces reward variance, accelerates convergence, and outperforms existing offline and online RL baselines.

## 2 Related Work

**Natural Language Agent Benchmark**    Recent studies have introduced evaluation tasks for language agents requiring long-horizon planning and strategic reasoning in multi-turn, goal-driven settings, including social conversations [25], strategy games (*e.g.*, *Werewolf*[26], *Avalon*[27]), economics-based scenarios (*e.g.*, *bargaining*[18; 19], *negotiation*[19]), and text-based games (*e.g.*, *Taboo*[28], *Guess My City*[8], *20 Questions*[8], *Ask-Guess*[29]). In this work, we focus on text-based games (*Guess My City*, *20 Questions*) and adversarial tasks (*Bargaining*, *Negotiation*). These settings require dynamic strategy adaptation, balancing short- and long-term goals, and complex reasoning, offering challenging benchmarks for evaluating LLM agents' planning and decision-making.

**Semantic Clustering**    Semantic clustering partitions samples into categories based on semantic similarity, typically by first extracting representations (*e.g.*, embeddings), then applying clustering algorithms such as $k$-means [30], hierarchical clustering [24], or DBSCAN [31]. In ARIA, actions

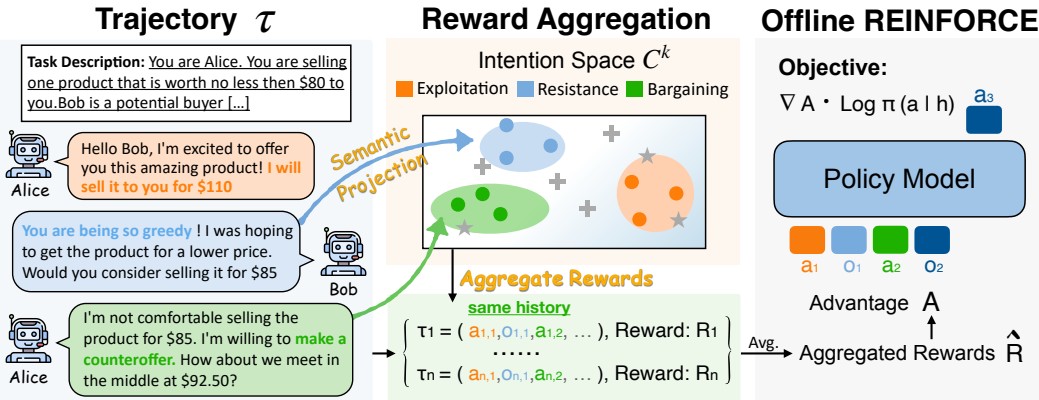

Figure 1: Illustration of ARIA. ARIA first lets agents interact to collect trajectories. Then it performs semantic projection and aggregates reward in the intention space, and finally updates the policy using the aggregated rewards.

are embedded and clustered into intentions using hierarchical clustering, which offers flexible post hoc granularity control and captures hierarchical semantic relations for coarse-to-fine strategy modeling.

**Training Language Agent with Reinforcement Learning**  Language agents often face ambiguous goals and sparse rewards, requiring adaptive long-term planning [1; 2; 3], which challenges decision-making. Reinforcement learning (RL) provides a principled framework to address these challenges, with existing methods falling into two categories: offline methods [12; 28; 13; 14; 26; 32], which pre-collect trajectories and apply post-processing (e.g., DPO [33], KTO [34]); and online methods [22; 20; 35; 36], which alternate between sampling and policy updates. However, the high-dimensional action space in free-form language tasks exacerbates reward sparsity and variance, hindering RL training. To mitigate this, we adopt an offline RL setup with reward aggregation and REINFORCE [23], improving learning stability and efficiency.

## 3 Method

We present an overview of ARIA in Figure 1. First, we construct the intention space using semantic clustering (§3.2), where the optimal granularity is determined by Reward-Oriented Granularity Selection (§3.4). Next, high-dimensional actions and observations are projected into the intention space through **semantic projection**, enabling reward aggregation (§3.3). Finally, the aggregated rewards are used to optimize the policy efficiently via offline REINFORCE (§3.5).

### 3.1 Task Formulation

In this paper, we select two types of open-ended language action tasks, single-agent and two-agent adversarial games, as the testbed. We formulate the tasks as a partially observable Markov decision process (POMDP) $\mathcal{M} = (\mathcal{S}, \mathcal{A}, \mathcal{O}, \mathcal{T}, \mathcal{R}, \gamma)$, where $\mathcal{S}$ is the global state, $\mathcal{A}$ is the action space of natural language actions, $\mathcal{O}$ is the observation, $\mathcal{T}$ is the transition function, $\mathcal{R}$ is the reward function, and $\gamma$ is the discount factor. In the **single-agent** setting, an agent $\mathcal{P}$ interacts with the environment by performing actions over time. At each step $t$, the agent receives an observation $o_t$ under state $s_t$ and maintains a history $h_t = \{o_1, a_1, \ldots, o_{t-1}, a_{t-1}, o_t\}$. The agent then selects an action $a_t \sim \pi_\theta(\cdot \mid h_t)$ conditioned on this history. The state $s_t$ subsequently transitions to $s_{t+1}$ according to the transition function $\mathcal{T} : \mathcal{S} \times \mathcal{A} \to \mathcal{S}$. When $s_t$ reaches the terminal condition, the environment returns a reward $\mathcal{R}$. The objective of the agent is to maximize the expected cumulative reward at the end of the episode based on the policy $\pi_\theta$. In the **adversarial** setting, two players $\mathcal{P} \in \{\mathcal{P}_1, \mathcal{P}_2\}$ take turns performing actions. In state $s_t$, player $\mathcal{P}_i$ selects an action $a_i \sim \pi_i(\cdot \mid h_t)$, where $h_t = \{o_1, a_1, \ldots, o_{t-1}, a_{t-1}, o_t\}$ is the history of observations and actions, and $o_t$ is derived from the state $s_t$ and the opponent's action $a_t$. The state $s_t$ then transitions to $s_{t+1}$ according to the transition function $\mathcal{T} : \mathcal{S} \times \mathcal{A} \to \mathcal{S}$. When the terminal condition is met in $s_t$, the environment

returns a reward $R$ to each player. Each player $\mathcal{P}_i$ aims to maximize the expected reward by the end of the episode based on their policy $\pi_i$.

## 3.2 Intention Space Construction

We construct a latent intention space using clustering. Given the action space $\mathcal{A}$ and observation space $\mathcal{O}$, each element $x \in \mathcal{A} \cup \mathcal{O}$ is embedded into a semantic vector using a pre-trained encoder $\phi : \mathcal{A} \cup \mathcal{O} \rightarrow \mathbb{R}^d$. We apply hierarchical agglomerative clustering [37] to partition the embedding space into $k$ clusters, forming the intention space $\mathcal{C}^k$ (see Appendix D for details). The number of clusters $k$ is selected via reward-oriented granularity selection (§3.4).

## 3.3 Reward Aggregation

Based on the intention space $\mathcal{C}^k$, we define a clustering function $c_k : \mathcal{A} \cup \mathcal{O} \rightarrow [k]$ that maps each element to a cluster index. At each step $t$, the action and observation are mapped to cluster labels $\tilde{a}_t = c_k(a_t)$ and $\tilde{o}_t = c_k(o_t)$, respectively. Given the history $h_t = \{a_1, o_1, \ldots, a_{t-1}, o_{t-1}\}$, the corresponding label sequence is

$$\tilde{h}_t = \{c_k(a_1), c_k(o_1), \ldots, c_k(a_{t-1}), c_k(o_{t-1})\}.$$

We aggregate rewards across history-action pairs that share the same semantic intention. The trajectory reward $R$ is assigned to intermediate steps using temporal discounting: $R(h_t, a_t) = \gamma^{T-t} R$, where $\gamma$ is the discount factor. For each intention pair $(\tilde{h}, \tilde{a})$, we compute the aggregated return by averaging over all history-action pairs that map to it:

$$\tilde{R}^{(k)}(\tilde{h}, \tilde{a}) = \frac{1}{|\mathcal{S}_{\tilde{h}, \tilde{a}}|} \sum_{(h_t, a_t) \in \mathcal{S}_{\tilde{h}, \tilde{a}}} R(h_t, a_t),$$

where $\mathcal{S}_{\tilde{h}, \tilde{a}} = \{(h_t, a_t) : c_k(h_t) = \tilde{h}, \; c_k(a_t) = \tilde{a}\}$ denotes the set of history-action pairs associated with intention $(\tilde{h}, \tilde{a})$. The aggregated return $\tilde{R}^{(k)}(\tilde{h}_t, \tilde{a}_t)$ is used as the advantage estimate $\tilde{A}(h_t, a_t)$ for policy optimization.

## 3.4 Reward-Oriented Granularity Selection

Semantic clustering helps compress the free-form, unstructured space of natural language actions and observations. However, selecting the appropriate granularity $k$ remains challenging. For example, in the context of negotiation, we compute standard clustering metrics—Silhouette Score [38], Calinski–Harabasz Index [39], and Davies–Bouldin Index [40]—across different configurations. In Figure 2, these metrics tend to favor overly coarse groupings due to the high similarity among actions, overlooking fine-grained distinctions that are critical for our task (see details of metric calculations in Appendix E).

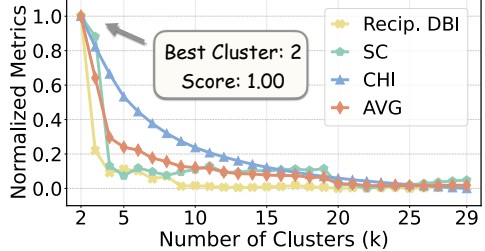

Figure 2: Clustering quality measured by SC, CHI, the reciprocal DBI and the average of three metrics. After normalization and averaging, $k = 2$ achieves the highest overall score.

To address this, we propose a reward-oriented granularity selection mechanism that assesses whether further splitting clusters yields meaningful reward change. Unlike traditional metrics based on geometric structure (*i.e.*, distance in embedding space), our method aligns with the RL objective by directly evaluating the impact on reward aggregation.

***SplitScore*** Let $k \in [2, K]$ denote all possible granularity levels. We use *SplitScore* to select the optimal granularity $k^*$, defined as *SplitScore*$(k) = \frac{\delta_k}{|\mathcal{D}|}$, where $\delta_k = \sum_{(h_t, a_t) \in \mathcal{D}} \left| \tilde{R}^{(k+1)}(h_t, a_t) - \tilde{R}^{(k)}(h_t, a_t) \right|$ represents the reward change for all $(h_t, a_t)$ pairs when the number of clusters changes from $k$ to $k+1$, and $\mathcal{D}$ is the collection of all $(h_t, a_t)$ pairs.

**Automatic Stopping Criterion**   To select the optimal granularity $k^*$, we define an early stopping mechanism based on *SplitScore*. Given a threshold $\epsilon > 0$ and a window size $\tau$[3], we stop splitting when $SplitScore(j) < \epsilon$ for all $j \in [k, k + \tau]$ as $k$ increases. The selected $k$ is then taken as $k^*$. We prove in Appendix C that *SplitScore* is bounded above a monotonically decreasing function. When *SplitScore* remains below the threshold, further splitting has minimal impact on $\delta_k$, indicating that the rewards $\tilde{R}^{(k)}(h_t, a_t)$ are nearly unchanged and do not significantly affect the training process. Thus, we select the smallest $k$ that meets the stopping condition to realize better space compression.

## 3.5   Offline REINFORCE with Aggregated Reward

We use the offline REINFORCE algorithm [23] to optimize the policy. Formally, let $\pi_\theta(a \mid s)$ denote the policy parameterized by $\theta$ and assign the aggregated reward $\tilde{R}^{(k)}(\tilde{h}_t, \tilde{a}_t)$ to $\tilde{A}(h_t, a_t)$. ARIA optimizes the model by maximizing the following objective:

$$J(\theta) = \mathbb{E}_{\tau \sim \pi_\theta} \left[ \sum_{t=0}^{T} \log \pi_\theta(a_t \mid h_t) \cdot \tilde{A}(h_t, a_t) \right].$$

# 4   Theoretical Analysis

In this section, we theoretically show that intention clustering-based aggregation of the rewards in ARIA can reduce the variance of the gradient descent while maintaining a small bound of bias, thus improving training stability and efficiency.

## 4.1   Background

Let $A(h_t, a_t)$ be the original advantage of $(h_t, a_t)$ and $c_k(x)$ be the cluster label assigned to instance $x \in \mathcal{A} \cup \mathcal{O}$ under the granularity $k$, we define $(\tilde{h}_t, \tilde{a}_t) = \{(c_k(a_1), c_k(o_1), \ldots, c_k(a_{t-1}), c_k(o_{t-1}), c_k(a_t)\}$ and calculate the cluster-averaged reward for $(\tilde{h}_t, \tilde{a}_t)$ as $\tilde{R}(\tilde{h}_t, \tilde{a}_t) = \frac{1}{|\mathcal{D}|} \sum_{(\tilde{h}_t, \tilde{a}_t) \in \mathcal{D}} R(\tilde{h}_t, \tilde{a}_t)$, where $R(\tilde{h}_t, \tilde{a}_t)$ is the original reward of $(\tilde{h}_t, \tilde{a}_t)$. Then we assign $\tilde{R}(\tilde{h}_t, \tilde{a}_t)$ to the advantage of $(h_t, a_t)$ as $\tilde{A}(h_t, a_t)$.

## 4.2   Main Theorem

We first establish that cluster-based aggregation reduces both the total variance of the policy gradient algorithm and the variance of the policy gradient. We give the following two lemmas.

**Lemma 4.1.** *Let $\tilde{A}$ denote the aggregated advantage, then $\mathrm{Var}(\tilde{A}) \leq \mathrm{Var}(A)$.*

**Lemma 4.2.** *Given the single-sample policy gradient estimator $\nabla_\theta \log \pi_\theta(a \mid h) A(h, a)$, the variance is reduced when using the aggregated advantage $\tilde{A}$. Specifically, $\mathrm{Var}(\nabla_\theta \log \pi_\theta \cdot \tilde{A}) \leq \mathrm{Var}(\nabla_\theta \log \pi_\theta \cdot A)$.*

We leave the proof in Appendix G. Building on Lemma 4.2, we show that the variance reduction by aggregation improves the convergence properties of offline REINFORCE.

**Theorem 4.1** (Variance-Improved Convergence)**.** *Given $N$ i.i.d. trajectories in train set, let $\hat{g} = \frac{1}{N} \sum_{i=1}^{N} \sum_t \nabla_\theta \log \pi_\theta(a_t^i \mid h_t^i) \tilde{A}_t^i$ be an estimator of the true gradient $g$. Define $\sigma^2 = \mathrm{Var}(\nabla_\theta \log \pi_\theta \cdot \tilde{A})$. Then, we have $\|\hat{g} - g\|_2 = O\left(\frac{\sigma}{\sqrt{N}}\right)$.*

*Proof.* Let $g = \mathbb{E}[g_i]$ be the expected gradient for the $i$-th trajectory, where $g_i = \sum_t \nabla_\theta \log \pi_\theta(a_t^i \mid h_t^i) \tilde{A}_t^i$ is the gradient estimator. The empirical gradient is $\hat{g} = \frac{1}{N} \sum_{i=1}^{N} g_i$. Let $\sigma^2 = \mathrm{Var}\left(\sum_t \nabla_\theta \log \pi_\theta(a_t \mid h_t) \cdot \tilde{A}_t\right)$. By expectation linearity and trajectory independence, the variance of the empirical gradient is $\mathbb{E}\left[\|\hat{g} - g\|_2^2\right] = \frac{\sigma^2}{N}$. By Jensen's inequality [41], we get $\mathbb{E}\left[\|\hat{g} - g\|_2\right] \leq \frac{\sigma}{\sqrt{N}}$. $\qquad \square$

---

[3]In this paper, we set $\epsilon = 0.01$ and $\tau = 10$. Ablation study on $\epsilon$ is provided in Appendix J.

Intuitively, because clustering reduces $\sigma$, supposing we want $|\hat{g} - g| < \epsilon$, convergence to within $\epsilon$ requires fewer samples, or equivalently, enables the use of larger step sizes for the same error tolerance. We then analyze the bias introduced by reward aggregation. To formalize this, we first give the notion of $\varepsilon$-bisimulation.

**Definition 1** ($\varepsilon$-Bisimulation). *Actions $a, a'$ are said to be $\varepsilon$-bisimilar if, for all states $s$, $|r(h, a) - r(h, a')| \leq \varepsilon$, $\quad D_{\mathrm{TV}}\big(P(\cdot \mid h, a), P(\cdot \mid h, a')\big) \leq \varepsilon$, where the total variation divergence $D_{\mathrm{TV}}\big(P(\cdot \mid h, a), P(\cdot \mid h, a')\big)$ measures how different the two distributions are over next states when different actions $a$ and $a'$ are taken at the same history $h$.*

**Theorem 4.2** (Bounded Bias via $\varepsilon$-Bisimulation). *Suppose the actions in each cluster are $\varepsilon$-bisimilar. Then, $\left| \mathbb{E}\left[ \nabla_\theta \log \pi_\theta(a \mid h)(A(h, a) - \tilde{A}(h, a)) \right] \right| \leq O(\varepsilon)$.*

*Proof.* $\varepsilon$-bisimulation ensures that value differences within a cluster satisfy $|Q^\pi(h, a) - Q^\pi(h, a')| \leq \frac{2\varepsilon}{1-\gamma}$, implying that cluster means differ by at most $O(\varepsilon)$. Since $\nabla \log \pi$ is bounded, the inner product bias is $O(\varepsilon)$. $\qquad\square$

In summary, by using conditional expectations and variance decomposition, we prove that replacing original advantages $A$ with cluster-averaged advantages $\tilde{A}$ removes the intra-cluster variance $\mathbb{E}[\mathrm{Var}(A \mid C)]$, lowering the total variance of the policy gradient estimate. Provided that the expectation remains approximately unchanged, this variance reduction leads to more stable training and faster convergence. It allows larger optimization steps without divergence and increases the utility of each sample, explaining why cluster-smoothed advantages yield smoother learning curves.

## 5 Experiments

### 5.1 Experimental Setup

**Baselines** We select both **online** and **offline** methods as baselines. For **offline methods**, we include: *1)* Behavior Cloning (BC) that trains the policy using successful trajectories. *2)* Trajectory-wise DPO [12], which trains langugae models using successful and failed trajectories. *3)* Step-wise DPO [13], which employs success/failure labels at the action level based on simulation outcomes. *4)* SPAG [28], which designs a discounted reward and uses offline PPO [22] for optimization of policy gradients. For **online methods**, we select: *1)* Archer [20], which utilizes a hierarchical reinforcement learning framework. *2)* StarPO [42], which applies GRPO [35] for policy optimization. Implementation details of baselines are in Appendix I.1.

**Tasks** We evaluate ARIA in both single-agent and adversarial environments (see Appendix H for details). For the **single-agent** setting, we consider two tasks: *1)* **Twenty Questions** [8], a dialogue task where the agent plays the role of a guesser, aiming to identify a hidden word selected from a list of 157 candidates by asking up to twenty yes-no questions. The Oracle responds with "Yes" "No" or "Invalid Question". The agent receives a final reward $\mathcal{R} = 1$ upon correctly guessing the target word, ending the episode; otherwise, the reward remains 0. *2)* **Guess My City** [8], a similar multi-turn task where the agent tries to identify a hidden city from a list of 100 candidates within twenty questions. The agent can ask any type of question and receives free-form responses, not limited to yes/no answers. For the **adversarial** setting, we consider two competitive tasks: *1)* **Bargaining** [43], a two-player game where Alice and Bob take turns proposing how to divide a fixed amount of money over a finite time horizon. As the game progresses, each player's payoff is discounted by a player-specific discount factor. If the game ends without an agreement, both players receive zero payoff. Otherwise, the discounted payoffs for Alice and Bob are given by $p_A$ and $p_B$. *2)* **Negotiation** [43], a two-player task where a seller (Alice) and a buyer (Bob) negotiate the price of a product with a true value. Alice and Bob each have subjective valuations. Over a fixed time horizon, the players alternate offers: at odd stages, Alice proposes a price and Bob decides whether to accept; at even stages, Bob proposes and Alice decides. If a price is accepted, the utilities for Alice and Bob are given by $u_A$, $u_B$. If no agreement is reached, both receive zero utility.

**Evaluation** For the **single-agent** environments, following ArCHer [20], we evaluate ARIA on a subset of $N$ tasks from Twenty Questions and Guess My City. We report the **average final reward**, defined as $\frac{1}{N} \sum_{i=1}^{N} \mathbb{I}[R_i = 1]$, where $R_i$ denotes the final reward for the $i$-th trajectory. We set $N = 200$ for each environment. For the **adversarial** environments, following GLEE [43], we evaluate ARIA across 48 game configurations. In each configuration, the agent plays as either Alice or Bob against fixed opponents, with each setting repeated $N = 25$ times. In Bargaining, the goal is to achieve a higher payoff than the opponent. In Negotiation, the objective is to sell at a higher price

Table 1: Main results on adversarial games. The best results are **bolded**, and the second best ones are underlined. The metric is the average win rate.

| Methods | Bargaining | | | | Negotiation | | | |
|---|---|---|---|---|---|---|---|---|
| | GPT-4o | Deepseek-V3 | Claude-3.5 | AVG. | GPT-4o | Deepseek-V3 | Claude-3.5 | AVG. |
| Vanilla Model | 30.14 | 24.05 | 33.72 | 29.30 | 37.92 | 36.94 | 40.08 | 38.31 |
| *Offline Baselines* | | | | | | | | |
| BC | 46.92 | 40.64 | 55.64 | 47.73 | 31.92 | 40.06 | 32.34 | 34.77 |
| Traj-wise DPO | 46.77 | 45.58 | 47.57 | 46.64 | 35.57 | 35.68 | 35.38 | 35.54 |
| Step-wise DPO | 48.91 | 55.48 | 46.00 | 50.13 | 36.33 | 41.56 | 49.17 | 42.35 |
| SPAG | 30.68 | 37.26 | 22.43 | 30.12 | 25.83 | 33.86 | 33.65 | 31.11 |
| *Online Baselines* | | | | | | | | |
| ArCHer | 43.78 | 47.35 | 53.94 | 48.36 | 35.00 | 37.84 | 34.64 | 35.83 |
| StarPO | 33.24 | 28.77 | 42.63 | 34.88 | 38.55 | 36.00 | 43.87 | 39.47 |
| *Ours* | | | | | | | | |
| ARIA ( *Iter 1*) | 51.54 | 55.26 | 52.66 | 53.15 | 45.65 | 42.69 | 49.02 | 45.79 |
| ARIA ( *Iter 2*) | 53.60 | **67.33** | 55.62 | **58.85** | **47.46** | 45.08 | 48.93 | 47.16 |
| ARIA ( *Iter 3*) | **58.66** | 55.83 | **59.55** | 58.01 | 46.48 | **50.50** | 49.42 | **48.80** |

(as the seller) or buy at a lower price (as the buyer). We let ARIA play both roles (Alice and Bob) against various opponents and compute the average win rate for each role, counting each successful completion of the task objective as a win. Specifically, the **average win rate** for Alice in Bargaining is defined as $W_A = \frac{1}{N} \sum_{i=1}^{N} \mathbb{I}[p_{i,A} > p_{i,B}]$, where $p_A$ and $p_B$ denote the discounted payoffs for Alice and Bob, respectively. The definition is symmetric for Bob. For Negotiation, the average win rate for alice is defined as $W_A = \frac{1}{N} \sum_{i=1}^{N} \mathbb{I}[u_{i,A} > u_{i,B}]$, where $u_A$ and $u_B$ represent the utilities of Alice and Bob. This is again symmetric for Bob.

**Models**    We use Llama-3-8B-Instruct [44] as the policy model. For each language action, we obtain its semantic embedding using text-embedding-3-small [45]. Additional ablation results on alternative embedding models are provided in Appendix K. In single-agent environments, Oracle is simulated with GPT-4. In adversarial settings, we employ opponent models from different families, including GPT-4o (gpt-4o-2024-08-06) [46], Claude 3 (claude-3-5-sonnet-20240620) [47], and DeepSeek-Chat (DeepSeek-V3) [48].

**Implementation Details**    For each scenario, we gather 1,000 games and update the policy using the trajectories. Specifically, in single-agent scenarios, the actor interacts directly with the Oracle (*i.e.*, the environment). For adversarial scenarios, we employ self-play to collect competitive interaction data from both players. To evaluate whether ARIA can consistently improve the policy, we perform three iterations. In each iteration, we collect another 1,000 games using the updated policy and conduct a new round of training. Additional implementation details are provided in Appendix I.

## 5.2   Results

**ARIA significantly improves policy performance.**    As shown in Table 1, in the adversarial tasks, ARIA achieves the highest average win rate in both *Bargaining* and *Negotiation*, surpassing offline and online baselines by 9.67% and 9.83%, respectively. Similarly, in the single-agent tasks (Table 2), ARIA outperforms all baselines by an average of 9.82%. Existing offline and online RL methods both rely on action sampling and reward assignment, where agents interact with the environment, collect action samples, and assign rewards to those actions. This approach works reasonably well in small action spaces, where repeated sampling provides stable and accurate reward estimates. However, in open-ended language action tasks, where agents act through natural language, the action space

Table 2: Main results on single-agent games. The best results are **bolded**, and the second-best ones are underlined. The metric is the average reward.

| Methods | Twenty. | Guess. | AVG. |
|---|---|---|---|
| Vanilla Model | 27.50 | 13.50 | 20.50 |
| *Offline Baselines* | | | |
| BC | 27.50 | 5.50 | 16.50 |
| Traj-wise DPO | 27.00 | 17.50 | 22.25 |
| Step-wise DPO | 27.50 | 11.50 | 19.50 |
| SPAG | 26.50 | 13.00 | 19.75 |
| *Online Baselines* | | | |
| ArCHer | 26.00 | 10.00 | 16.25 |
| StarPO | 27.50 | 10.50 | 16.00 |
| *Ours* | | | |
| ARIA *(Iter 1)* | 28.00 | 29.00 | 28.50 |
| ARIA *(Iter 2)* | 29.50 | 32.00 | 30.75 |
| ARIA *(Iter 3)* | **34.50** | **36.00** | **35.25** |

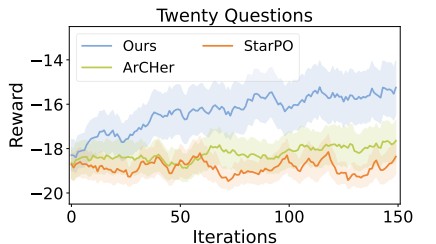 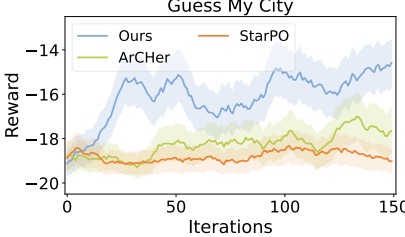

(a) Reward over Iterations on *Twenty Ques-* (b) Reward over Iterations on *Guess My City*.
*tions*.

Figure 3: (a) and (b) show the reward curves of ARIA and other online methods over iterations on the *Twenty Questions* and *Guess My City* respectively.

grows exponentially to $\mathcal{V}^{\mathcal{L}}$, given a vocabulary of size $\mathcal{V}$ and an average sequence length $\mathcal{L}$. In such vast spaces, each sample typically receives only a binary reward signal, and the sample size $\mathcal{N}$ is much smaller than the action space, leading to highly sparse and noisy reward signals and making accurate credit assignment challenging. ARIA addresses this by introducing reward aggregation in the intention space, which reduces reward variance and significantly improves learning performance.

**ARIA continuously improves policy through iteration.**    After confirming that ARIA significantly outperforms the baselines, we further investigate its performance under iterative updates. As shown in Table 1 and Table 2, ARIA achieves additional gains of 3.27% and 1.85% after two and three iterations, respectively. This suggests that reward aggregation effectively reduces variance while preserving essential discriminative signals for policy learning, reflecting a favorable bias-variance trade-off. It further enhances sample efficiency and mitigates the risk of premature convergence caused by excessive smoothing, demonstrating that reward aggregation can deliver stable and cumulative performance improvements.

## 5.3    Extending to Online ARIA

**Settings**    We first perform reward aggregation using pre-collected trajectories. The aggregated rewards are then used to initialize a point-wise reward model (RM), implemented as Llama-3.1-8B-Instruct [44], consistent with the policy model. Subsequently, the policy interacts with the environment to dynamically generate new samples, which are scored by the RM to update the policy. Additionally, the RM is periodically updated with the latest collected data, allowing it to evolve alongside the policy. We conduct the online ARIA on two single-agent games to conveniently observe reward at each iteration. Detailed parameter settings are provided in Appendix I.2.

**Results**    As shown in Figure 3, ARIA achieves faster reward improvement and consistently higher returns across iterations compared to existing online methods (ArCHer and StarPO). This improvement stems from two key advantages: *1)* Reward aggregation provides an initial dense and low-variance reward signal, accelerating early-stage policy learning. *2)* The dynamic RM update ensures alignment between the reward function and the evolving policy, preventing drift and reward misalignment common in static settings. Together, these factors enhance both sample efficiency and reward shaping accuracy, leading to faster and more stable policy improvement.

## 6    Analysis

### 6.1    Reward Aggregation Significantly Reduces Reward Variance

We show variance change before and after reward aggregation in Figure 4. As shown in Figure 4a, reward aggregation markedly reduces the fluctuation range of action rewards. The original binary reward distributions are highly polarized, with values mostly concentrated near 0 or 1. In a large action space, most actions are sampled only once, and the corresponding binary reward is directly assigned to each action, resulting in high reward variance. By contrast, after reward aggregation, actions within the same cluster share a common reward, which significantly smooths the distribution and reduces variance. Figure 4b further demonstrates that reward variance decreases across all four tasks, highlighting the effectiveness and necessity of reward aggregation in stabilizing policy learning.

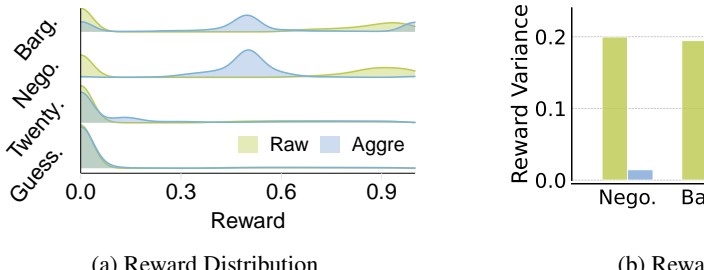

(a) Reward Distribution  (b) Reward Variance

Figure 4: (a) illustrates the distribution of rewards. (b) presents the change in reward variance.

## 6.2 Reward Aggregation Improves Policy Optimization

To evaluate whether reward aggregation improves training efficiency, we first compare the policy loss curves under different reward shaping strategies in Figure 5b. The results show that ARIA, which applies semantic-level reward aggregation, accelerates loss reduction compared to the vanilla REINFORCE baseline. This indicates that shaping the reward through aggregation provides a stronger learning signal, enabling faster policy updates and improved sample efficiency in offline training. We further observe that, despite converging to similar loss levels, the methods exhibit substantial differences in downstream performance. As shown in Figure 5a, ARIA outperforms other variants by 17.91% and 13.80% on the bargaining and negotiation tasks, respectively. We attribute these gains to the complementary effects of reward decay and reward aggregation: Reward decay introduces temporal structure that helps assign credit to early-stage actions, but plays a limited role in reducing signal noise. In contrast, reward aggregation substantially lowers reward variance by assigning shared signals to semantically similar actions, thereby improving the quality of gradient estimation. This variance reduction enables more stable and efficient optimization and plays a central role in enhancing policy performance in open-ended language action settings.

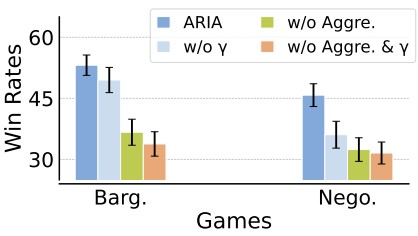

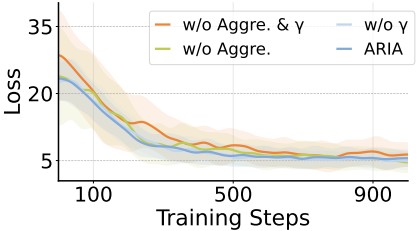

(a) Ablation of ARIA on win rates  (b) Ablation of ARIA on loss (smoothed)

Figure 5: Ablation of ARIA. (a) shows win rates on adversarial games and (b) shows training loss curves under different ablation settings in adversarial games.

## 6.3 Generalization of ARIA to Other Models

Table 3: ARIA on Qwen2.5-7B-Instruct and Qwen2.5-1.5B-Instruct.

| Methods | Bargaining | Negotiation | AVG. |
|---------|-----------|-------------|------|
| *Qwen2.5-7B-Instruct* | | | |
| Vanilla | 37.92 | 35.50 | 36.71 |
| ARIA | 65.96 | 47.06 | 56.51 (+19.8 ↑) |
| *Qwen2.5-1.5B-Instruct* | | | |
| Vanilla | 0.02 | 18.22 | 9.12 |
| ARIA | 0.01 | 20.47 | 10.24 (+1.12 ↑) |

In Section 5.2, we show that ARIA achieves significant improvements on Llama3-8B-Instruct. To further assess the transferability of ARIA, we apply it to the Qwen models (Qwen2.5-7B-Instruct [49] and Qwen2.5-1.5B-Instruct [49]) and conduct comparative experiments on two adversarial games[4]. As shown in Table 3, we observe that altering the base model consistently yields improvements. This

---

[4]All the settings are the same as those in Section 5.

suggests that our reward aggregation approach is model-agnostic and independent of specific architectural features or pretraining data of the underlying language models. We attribute this generalizability to the shared structural properties in the semantic spaces learned by large-scale language models. By performing aggregation in the intention space, ARIA leverages these commonalities to reduce reward variance while preserving task-specific discriminative signals.

# 7    Conclusion

In this paper, we address the core challenges of reinforcement learning in open-ended language action tasks, where agents must operate in **exponentially large action spaces and learn from sparse, delayed rewards**. To tackle the resulting high variance in policy optimization, we introduce **semantic projection**, a novel intention-aware framework that maps natural language actions from the high-dimensional token space into a low-dimensional intention space. This projection enables reward aggregation across semantically similar actions, effectively densifying sparse rewards and reducing gradient variance. Built on this idea, we propose ARIA, which automatically discovers task-specific intention structures via hierarchical clustering and integrates the aggregated rewards into REINFORCE for efficient policy learning. We further provide a theoretical analysis showing that replacing original advantages with cluster-averaged advantages reduces intra-cluster variance, thereby lowering the overall variance of the policy gradient and improving learning stability. Extensive experiments across four diverse tasks—including both single-agent and adversarial two-agent games—demonstrate that ARIA improves training stability, accelerates convergence, and consistently outperforms strong offline and online RL baselines. Our findings highlight the importance of structure-aware reward shaping in scaling reinforcement learning for language agents in open-ended environments.

## Acknowledgement

We appreciate the support from the Chinese NSF General Program (No.62572129), Major Research Plan (No.92270121). We also acknowledge the use of an icon from Flaticon[5] and thank its creators for providing this visually appealing design.

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

# Appendix

## A  Limitations

While ARIA shows strong performance across various single-agent and adversarial tasks, it relies on clustering in the semantic embedding space to define intention groups, which introduces two limitations. First, the effectiveness of reward aggregation depends on the quality of sentence embeddings. If embeddings fail to capture fine-grained behavioral differences, clustering may become coarse or misaligned, impairing learning. Second, the current formulation assumes that intentions are discrete and well-separated. This assumption may not hold in tasks with overlapping goals. Extending ARIA to support soft or continuous intention representations and incorporating task-specific structures into the projection process are promising directions for future work.

## B  Broader Impacts

The contribution of our work lies in the proposed intention-aware reward aggregation framework, which demonstrates a principled and effective approach for training language agents in open-ended language action environments with sparse and delayed rewards. We focus on tasks such as negotiation, goal-oriented dialogue, and multi-turn interaction, as they reflect real-world scenarios that demand strategic reasoning and adaptive language generation. Compared to traditional structured tasks with predefined action spaces, these open-ended language interaction tasks better align with human communication dynamics and present a valuable testbed for exploring the cognitive and social capabilities of language agents.

Our method is not limited to the evaluated benchmarks (e.g., *Negotiation*, *Bargaining*, *20 Questions*, *Guess My City*), but can generalize to a broader range of domains involving multi-agent decision-making and goal-driven communication, such as collaborative problem-solving, strategic planning, and educational tutoring systems. By enhancing the sample efficiency and robustness of reinforcement learning for LLM-based agents, our framework contributes to the development of socially intelligent, general-purpose AI systems that can interact with humans in nuanced and adaptive ways.

## C   Analysis of SplitScore

In section 3.4, we claim that *SplitScore* is bounded above a monotonically decreasing function. When *SplitScore* consistently falls below a predefined threshold, it indicates that further splits contribute little to the total reward change $\delta_k$. In this case, the reward values $\tilde{R}(h_t, a_t)$ remain largely stable, and additional splits are unlikely to affect training outcomes significantly. We provide the detailed explanation as follows.

We begin by recalling the definition:

$$SplitScore(k) = \frac{\delta_k}{|\mathcal{D}|}, \tag{1}$$

where $\delta_k = \sum_{(h_t, a_t) \in \mathcal{D}} \left| \tilde{R}^{(k+1)}(h_t, a_t) - \tilde{R}^{(k)}(h_t, a_t) \right|$ represents the total absolute change in reward across all $(h_t, a_t)$ when the clustering granularity increases from $k$ to $k+1$. Here, $\mathcal{D}$ denotes the set of all action instances.

We can reformulate Equation 1 as

$$SplitScore(k) = \frac{n_k \cdot \bar{\delta}_k}{|\mathcal{D}|},$$

where $\mathcal{D}_k \subseteq \mathcal{D}$ is the set of instances affected by the change in clustering, $n_k = |\mathcal{D}_k|$, and $\bar{\delta}_k = \frac{1}{n_k} \sum_{(h_t, a_t) \in \mathcal{D}_k} \left| \tilde{R}^{(k+1)}(h_t, a_t) - \tilde{R}^{(k)}(h_t, a_t) \right|$ is the average reward change over the affected instances.

**Theoretical Boundaries and Edge Cases.**   Given the reward $\tilde{R}(h_t, a_t) \in [0, 1]$, it follows that $\bar{\delta}_k \in [0, 1]$. This leads to the following inequality:

$$0 \leq SplitScore(k) = \frac{n_k \cdot \bar{\delta}_k}{|\mathcal{D}|} \leq \frac{n_k}{|\mathcal{D}|} \leq \frac{n_{k,\max}}{|\mathcal{D}|},$$

where $n_k$ is the number of affected instances with $k$ clusters, and $n_{k,\max}$ denotes its maximum possible value. Since hierarchical clustering splits one cluster at a time, the number of affected instances $n_k$ typically decreases as $k$ increases. Therefore, $n_{k,\max}$ is a monotonically decreasing function of $k$, which ensures the convergence of *SplitScore*.

We further note two edge cases:

*1)* If $n_k = 0$ or $\bar{\delta}_k = 0$, then *SplitScore*$(k) = 0$, indicating that the split causes no reward change.
*2)* If $n_k = n_{k,\max}$ and $\bar{\delta}_k = 1$, the split results in the maximum possible total reward change.

Therefore, the decay of *SplitScore* provides a natural criterion for early stopping, as it reflects diminishing changes in the reward signal expressivity with respect to further semantic partitioning.

## D   Algorithm of Hierarchical Agglomerative Clustering

We illustrate the process of the Hierarchical Agglomerative Clustering (HAC) algorithm in Algorithm 1.

## E   Clustering Metric Calculation Details

We use three standard indicators to evaluate clustering performance: the Silhouette Coefficient [38], the Calinski-Harabasz Index [39], and the Davies-Bouldin Index [40].

### E.1   Silhouette Coefficient

The Silhouette Coefficient is a widely used metric for evaluating clustering quality. It captures two key aspects: cohesion, which measures how closely related the objects within a cluster are, and

---

**Algorithm 1** Hierarchical Agglomerative Clustering (HAC) with Average Linkage

---

**Require:** Dataset $\mathcal{X} = \{\boldsymbol{x}_1, \ldots, \boldsymbol{x}_n\}$
**Ensure:** A dendrogram representing the hierarchy of clusters
1: Initialize clusters: $\mathcal{C} \leftarrow \{\{\boldsymbol{x}_1\}, \ldots, \{\boldsymbol{x}_n\}\}$
2: **while** $|\mathcal{C}| > 1$ **do**
3:    Compute pairwise distances using average linkage:

$$(C_p, C_q) = \arg\min_{C_i \neq C_j \in \mathcal{C}} \frac{1}{|C_i||C_j|} \sum_{\boldsymbol{x} \in C_i} \sum_{\boldsymbol{y} \in C_j} \|\boldsymbol{x} - \boldsymbol{y}\|_2$$

4:    Merge clusters: $C_{\text{new}} \leftarrow C_p \cup C_q$
5:    Update cluster set:
$$\mathcal{C} \leftarrow (\mathcal{C} \setminus \{C_p, C_q\}) \cup \{C_{\text{new}}\}$$

6: **end while**
7: **return** Deprogram recording the merge steps

---

separation, which assesses how distinct a cluster is from others. For each sample $i$, the Silhouette Coefficient $s(i)$ is defined as

$$s(i) = \frac{b(i) - a(i)}{\max\{a(i), b(i)\}},$$

where $a(i)$ denotes the average distance between $i$ and all other points in the same cluster (intra-cluster distance), and $b(i)$ is the minimum average distance from $i$ to all points in any other cluster, of which $i$ is not a member (nearest-cluster distance).

The value of $s(i)$ ranges from $-1$ to $1$. A value close to $1$ indicates that the sample is well matched to its own cluster and poorly matched to neighboring clusters. A value near $0$ suggests that the sample lies between two clusters. A negative value implies potential misclassification, where the sample may have been assigned to the wrong cluster. The overall quality of a clustering configuration can be quantified by the mean Silhouette Coefficient across all samples.

## E.2   Calinski-Harabasz Index

The Calinski-Harabasz Index (CHI) evaluates clustering quality based on the principle that good clusters should be compact and well separated. Given a clustering result with $k$ clusters and $n$ total samples, CHI is defined as

$$\text{CHI} = \frac{\text{Tr}(B_k)}{\text{Tr}(W_k)} \cdot \frac{n - k}{k - 1},$$

where $\text{Tr}(B_k)$ is the trace of the between-cluster dispersion matrix, which measures the distance of each cluster center from the overall mean, and $\text{Tr}(W_k)$ is the trace of the within-cluster dispersion matrix, indicating the compactness of each cluster.

A higher CHI value suggests better-defined clusters, with dense intra-cluster groupings and well-separated inter-cluster distances. This metric is particularly effective when the number of clusters $k$ is known or fixed.

## E.3   Davies-Bouldin Index

The Davies-Bouldin Index (DBI) is an internal metric for evaluating clustering quality. It measures the average similarity between each cluster and its most similar one, combining both intra-cluster compactness and inter-cluster separation. Given a clustering result with $k$ clusters, DBI is defined as

$$\text{DBI} = \frac{1}{k} \sum_{i=1}^{k} \max_{j \neq i} \left( \frac{S_i + S_j}{M_{ij}} \right),$$

where $S_i$ is the average distance between each point in cluster $i$ and its centroid (i.e., intra-cluster dispersion), and $M_{ij}$ is the distance between the centroids of clusters $i$ and $j$ (i.e., inter-cluster separation). The term inside the maximum quantifies the similarity between clusters $i$ and $j$.

A lower DBI indicates better clustering, as it reflects compact, well-separated clusters. This index is particularly useful for comparing the quality of different clustering results on the same dataset.

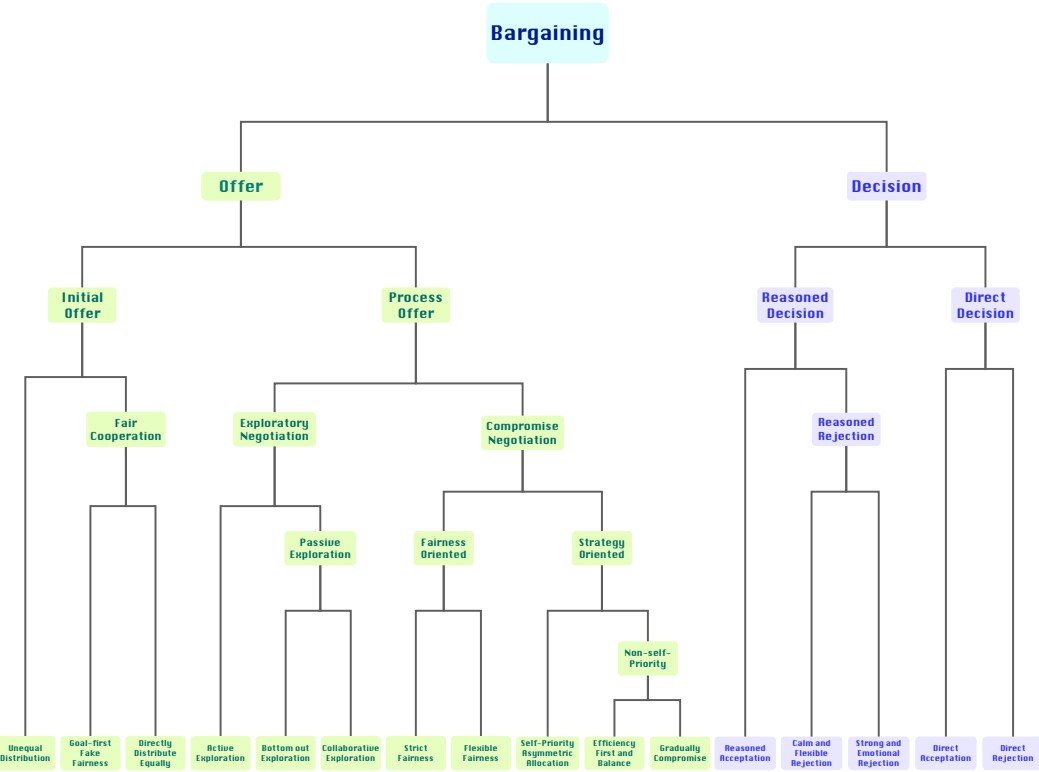

Figure 6: Tree-like clustering result example of bargaining.

## F  Illustration of Results after Semantic Projection.

We conduct a preliminary analysis of the action categories derived from semantic clustering. Specifically, we select 1,000 gameplay trajectories from Bargaining scenarios and apply hierarchical clustering on the extracted actions, setting the number of clusters $k = 16$. To gain deeper insights into the clustering structure and semantics of each category, we utilize GPT-4o to extract representative features from the actions within each cluster. This process allows us to identify shared characteristics within individual clusters and perform comparative analysis across clusters, thereby facilitating a comprehensive understanding of the entire hierarchical structure, as illustrated in Figure 6. Our analysis follows three main steps:

*1*) **Intra-cluster Feature Extraction**: For each of the 16 clusters, we input the corresponding actions into GPT-4o, leveraging its strong semantic reasoning capabilities to extract the common features. These distilled features serve as the basic descriptors for the leaf nodes in Figure 6.

*2*) **Comparative Analysis**: To refine these descriptors, we perform pairwise comparisons between sibling clusters that share the same parent node in the hierarchy. GPT-4o is used to analyze the semantic differences between such pairs, filtering out redundant or overlapping traits and preserving only the core distinguishing features.

*3*) **Hierarchical Backtracking and Merging**: After characterizing all leaf-level clusters, we recursively merge sibling nodes to form higher-level categories. At each level of merging, we repeat the previous two steps, feature extraction and comparative analysis, to summarize semantic attributes at internal nodes. This iterative bottom-up process enables us to construct a layered interpretation of the entire clustering tree.

As shown in Figure 6, at the top level, the actions are divided into two major phases: Offer and Decision, reflecting the progression of bargaining interactions. The Offer phase is further decomposed into subcategories such as Initial Offer, Exploratory Negotiation, and Compromise Negotiation, capturing different negotiation strategies ranging from fairness-oriented to strategically self-serving. The Decision phase includes Reasoned and Direct responses, distinguishing between deliberative and immediate choices.

# G Proof of Lemma

In this section, we give detailed proof of the Lemma in Section 4.

**Lemma G.1.** *Let $\tilde{A}$ denote the aggregated advantage, then* $\mathrm{Var}(\tilde{A}) \leq \mathrm{Var}(A)$.

*Proof.* Let $C$ denote the chosen cluster under granularity $k$. By the law of total variance, we have

$$\mathrm{Var}(A) = \mathbb{E}\left[\mathrm{Var}(A \mid C)\right] + \mathrm{Var}\left(\mathbb{E}[A \mid C]\right).$$

Since $\mathbb{E}_C\left[\mathrm{Var}(A \mid C)\right] \geq 0$, it follows that

$$\mathrm{Var}(\tilde{A}) = \mathrm{Var}\left(\mathbb{E}[A \mid C]\right) = \mathrm{Var}(A) - \mathbb{E}\left[\mathrm{Var}(A \mid C)\right] \leq \mathrm{Var}(A).$$

$\square$

Intuitively, replacing each trajectory's advantage with the cluster average filters out intra-cluster noise, leading to a more stable estimate. We then show that replacing the original advantage $A$ with the aggregated advantage $\tilde{A}$ reduces the variance of the policy gradient estimator.

**Lemma G.2.** *Given the single-sample policy gradient estimator* $\nabla_\theta \log \pi_\theta(a \mid h) A(h, a)$, *the variance is reduced when using the aggregated advantage* $\tilde{A}$. *Specifically,* $\mathrm{Var}(\nabla_\theta \log \pi_\theta \cdot \tilde{A}) \leq \mathrm{Var}(\nabla_\theta \log \pi_\theta \cdot A)$.

*Proof.* The variance of the single-sample policy gradient estimator can be written as

$$\mathrm{Var}(\nabla_\theta \log \pi_\theta \cdot A) = \mathbb{E}\left[(\nabla_\theta \log \pi_\theta)^2 A^2\right] - \left(\mathbb{E}[\nabla_\theta \log \pi_\theta \cdot A]\right)^2.$$

Replacing $A$ with a constant $\tilde{A}$ within each cluster leads to the following decomposition:

$$\mathbb{E}\left[(\nabla_\theta \log \pi_\theta)^2 A^2\right] - \mathbb{E}\left[(\nabla_\theta \log \pi_\theta)^2 \tilde{A}^2\right] = \mathbb{E}\left[\mathbb{E}\left[(\nabla_\theta \log \pi_\theta)^2 (A - \tilde{A})^2 \mid C\right]\right] \geq 0.$$

Therefore,

$$\mathrm{Var}(\nabla_\theta \log \pi_\theta \cdot \tilde{A}) \leq \mathrm{Var}(\nabla_\theta \log \pi_\theta \cdot A).$$

$\square$

# H   Task Details

**20 Questions (Twenty Questions) [8]**   This game evaluates an agent's ability to gather information and reason about an unknown object based on limited data. One participant (the oracle) selects an object, while the other (the guesser) attempts to identify it by asking a series of yes/no questions. In our setting, the GPT-4o serves as the oracle, and the agent's goal is to develop an effective questioning policy to identify the object within a fixed number of turns. This setup assesses both the agent's reasoning abilities and its semantic understanding of the objects involved.

**Guess My City [8]**   This more complex game involves two participants: the oracle, who is associated with a specific city, and the guesser, who attempts to determine the oracle's hometown. Unlike *20 Questions*, the guesser can pose both yes/no and open-ended questions, enabling richer and more informative exchanges. This task challenges the agent's strategic planning and language comprehension, requiring it to generate meaningful questions that elicit valuable clues and increase its likelihood of correctly identifying the city.

**Bargaining [43]**   This is a two-player game where Alice and Bob take turns proposing how to divide a fixed amount of money $M$ over a finite time horizon $T$. As the game progresses, each player's payoff is discounted by a player-specific discount factor, $\delta_A$ for Alice and $\delta_B$ for Bob. The outcome of the game is denoted by a pair $(t_{\text{ev}}, p_{\text{ev}})$, where $t_{\text{ev}}$ indicates the round at which the game terminates, and $p_{\text{ev}}$ represents the share of $M$ that Alice receives (before applying discounting). If the game ends without an agreement, we set $t_{\text{ev}} = \infty$, and both players receive zero payoff. Otherwise, the discounted payoffs are given by $p_A = \delta_A^{t_{\text{ev}}-1} p_{\text{ev}}$ and $p_B = \delta_B^{t_{\text{ev}}-1}(1 - p_{\text{ev}})$.

**Negotiation [43]**   This is a two-player task where a seller (Alice) and a buyer (Bob) negotiate the price of a product with a true value $V$. Alice and Bob each have subjective valuations, $V_A$ and $V_B$, respectively. Over a fixed time horizon $T$, the players alternate offers: at odd stages, Alice proposes a price and Bob decides whether to accept; at even stages, Bob proposes and Alice decides. If a price $p$ is accepted, the utilities are $u_A = p - V_A$ for Alice and $u_B = V_B - p$ for Bob. If no agreement is reached, both receive zero utility.

# I   Implementation details

## I.1   Baselines

To ensure a fair comparison, all methods are trained using the same amount of data. For offline methods, we collect 1,000 trajectories in the **single-agent** scenario and 2,000 trajectories in the **adversarial** scenario, corresponding to 1,000 games where both Alice and Bob contribute 1,000 trajectories each. Models are trained for three epochs on a combined dataset consisting of two tasks from the same category (single-agent or adversarial).

For online methods, we perform 150 iterations in both scenarios. In each iteration, we conduct 32 games in the single-agent setting and 32 self-play games in the adversarial setting. For ArCHer and online ARIA, the final reward of each collected trajectory is distributed across steps, and models are updated at the utterance level in each iteration. For RAGEN(GRPO), we group trajectories into four groups, compute the advantage for each group, and perform trajectory-level updates. All experiments are conducted using 8 NVIDIA A100-80GB GPUs.

## I.2   Parameter Design

As described in § 5.1, we train the actor model using both online and offline methods. We use the parameter efficient finetuning technique, specifically LoRA (Target $q_{proj}, k_{proj}, v_{proj}$, rank=8, $\alpha$=16). The hyperparameter configurations for all experiments are detailed in Table 4.

Table 4: Hyperparameters for All Experiments

| | | Adversarial | Single-Agent |
|---|---|---|---|
| BC | actor lr | 2e-5 | 2e-5 |
| | batch size | 32 | 16 |
| | number of epoch | 3 | 3 |
| | cutoff length | 4096 | 4096 |
| Trajectory-wise DPO | actor lr | 2e-5 | 2e-5 |
| | kl coefficient | 0.2 | 0.2 |
| | batch size | 16 | 16 |
| | number of epoch | 3 | 3 |
| | cutoff length | 4096 | 4096 |
| Step-wise DPO | actor lr | 2e-5 | 2e-5 |
| | batch size | 32 | 16 |
| | number of epoch | 3 | 3 |
| | cutoff length | 4096 | 4096 |
| SPAG | actor lr | 2e-5 | 2e-5 |
| | batch size | 32 | 16 |
| | number of epoch | 3 | 3 |
| | cutoff length | 4096 | 4096 |
| ArCHer | rollout trajectories | 32 | 32 |
| | replay buffer size | 10000 | 10000 |
| | actor lr | 3e-6 | 3e-6 |
| | critic lr | 6e-5 | 6e-5 |
| | batch size | 64 | 64 |
| | critic updates per iteration | 50 | 50 |
| | actor updates per iteration | 10 | 10 |
| | warm up iters with no actor update | 10 | 10 |
| | iteration | 150 | 150 |
| StarPO | rollout trajectories | 32 | 32 |
| | group size | 8 | 4 |
| | actor lr | 3e-6 | 3e-6 |
| | batch size | 32 | 32 |
| | iteration | 150 | 150 |
| ARIA (Offline) | actor lr | 2e-5 | 2e-5 |
| | batch size | 32 | 16 |
| | number of epoch | 3 | 3 |
| | cutoff length | 4096 | 4096 |
| ARIA (Online) | rollout trajectories | 32 | 32 |
| | actor lr | 3e-6 | 3e-6 |
| | batch size | 64 | 64 |
| | actor updates per iteration | 10 | 10 |
| | iteration | 150 | 150 |
| Reward Model | lr | 2e-5 | 2e-5 |
| | batch size | 64 | 64 |
| | number of epoch | 3 | 3 |
| | update | per 50 steps | per 50 steps |
| | cutoff length | 4096 | 4096 |

Table 7: The results of significant tests.

| Methods | Bargaining | | | Negotiation | | |
|---|---|---|---|---|---|---|
| | Win Rate | t-value | p-value | Win Rate | t-value | p-value |
| Vanilla Model | 29.30 | 15.53 | <0.001 | 38.31 | 5.92 | <0.001 |
| *Offline Baselines* | | | | | | |
| BC | 47.73 | 1.6711 | 0.0475 | 34.77 | 9.91 | <0.001 |
| Traj-wise DPO | 46.64 | 2.99 | 0.0013 | 35.54 | 9.60 | <0.001 |
| Step-wise DPO | 50.13 | 2.60 | 0.0047 | 42.35 | 3.00 | 0.0014 |
| SPAG | 30.12 | 14.56 | <0.001 | 31.11 | 12.67 | <0.001 |
| *Online Baselines* | | | | | | |
| ArCHer | 48.36 | 1.6463 | 0.0499 | 35.83 | 7.85 | <0.001 |
| StarPO | 34.88 | 11.5321 | <0.001 | 39.47 | 4.66 | <0.001 |
| *Ours* | | | | | | |
| ARIA | 53.15 | – | – | 45.79 | – | – |

## J  Ablation on the Threshold $\epsilon$

We conduct an ablation study to examine the effect of different thresholds $\epsilon$ for *SplitScore* on performance. Specifically, we compare $\epsilon = 0.1$, which corresponds to bargaining with $k = 4$ clusters and negotiation with $k = 2$, and $\epsilon = 0.001$, which corresponds to both bargaining and negotiation with

Table 5: Ablation of Threshold $\epsilon$.

| Methods | Bargaining | Negotiation | AVG. |
|---|---|---|---|
| ARIA ($\gamma = 0.01$) | **53.15** | **45.79** | **49.47** |
| w/ $\gamma = 0.1$ | 43.86 | 38.02 | 40.91 (-8.56 ↓) |
| w/ $\gamma = 0.001$ | 46.63 | 35.77 | 41.20 (-8.27 ↓) |

$k = 100$. As shown in Table 5, a larger $\epsilon$ results in coarser reward aggregation, potentially assigning the same reward to actions with different semantics, which degrades performance. Conversely, a smaller $\epsilon$ causes overly fine-grained aggregation, making the reward signal too sparse for effective learning, which also harms performance. Therefore, we set $\epsilon = 0.01$ for all experiments.

## K  Ablation on the Embedding Models

We perform ablation studies using embedding models with varying capabilities: text-embedding-3-small [45], Qwen3-Embedding-0.6B [50] (stronger), and e5-large-v2 [51] (weaker). As shown in Table 6, ARIA maintains strong performance with higher-quality embed-

Table 6: Ablation of Embedding Models.

| Model | Bargaining | Negotiation | AVG |
|---|---|---|---|
| text-embedding-3-small | 51.54 | 45.65 | 48.595 |
| Qwen3-Embedding | **53.30** | 43.10 | 48.20 |
| e5-large-v2 | 49.37 | 38.74 | 44.055 |

dings (*e.g.* text-embedding-3-small and Qwen3-Embedding), demonstrating robustness to different embedding choices. In contrast, performance degrades with the weaker e5-large-v2 model, highlighting the importance of embedding quality for capturing accurate semantic similarity.

## L  Statistical Significance of Experiments

We perform statistical significance testing to assess the effectiveness of ARIA compared to each baseline on two multi-agent tasks: Bargaining and Negotiation. For each baseline, we report the mean performance, the t-value, and the p-value from a paired t-test comparing ARIA against the baseline. As shown in Table 7, ARIA consistently outperforms all baselines across both tasks. The improvements are statistically significant ($p < 0.05$) in all cases, demonstrating that ARIA provides meaningful gains over existing offline and online approaches.

# M  Case Study

We evaluate the performance of agents trained by ARIA in both single-agent (*Twenty Questions*, *Guess My City*) and multi-agent (*Bargaining*, *Negotiation*) scenarios. In the single-agent tasks, the agent successfully completes *Twenty Questions* and *Guess My City* within 5 and 9 turns, respectively. For the multi-agent settings, the ARIA-trained agent plays the role of Bob, while Alice is simulated by GPT-4o. In both *Bargaining* and *Negotiation* tasks, the agent consistently adopts effective strategies to maximize its gains.

Listing 1: Case of Twenty Questions

```
Actor: Is it a living thing? Oracle: Yes.
Actor: Is it a mammal? Oracle: Yes.
Actor: Is it a human? Oracle: No.
Actor: Is it a carnivore? Oracle: No.
Actor: Is it a cow? Oracle: Yes.
```

Listing 2: Case of Guess My City

```
Actor: Is your city the capital of a country?
Oracle: Yes, it is located in the capital of a country.
Actor: Is the country you're thinking of located in Europe?
Oracle: No, the country is not located in Europe.
Actor: Is the country you're thinking of located in North America?
Oracle: No, it is not located in North America.
Actor: Is the country you're thinking of located in South America?
Oracle: No, the country is not located in South America.
Actor: Is the country you're thinking of located in Africa?
Oracle: No, the country is not located in Africa.
Actor: Is the country you're thinking of located in Asia?
Oracle: Yes, it is located in Asia.
Actor: Is your city located in China?
Oracle: No, it is not located in China.
Actor: Is your city located in the country of Japan?
Oracle: Yes, it is located in the country of Japan.
Actor: Is your city Tokyo?.
Oracle: Yes.
```

Listing 3: Case of Bargaining

Task Description:
In this game, there are two players. You are playing as Bob. You and Alice need to divide 100$ between you. In each round of the game, one of the players will make a proposal for division and can describe their proposal in a few words. The other player will decide whether to accept the offer and finish the game, or to proceed to the next round, in which they will be the one making the proposal. Your goal in this game is to achieve more money than Alice. Beware of inflation! With each passing round, the money is worth 0% less for you. For Alice, the money is worth 20% less. You have 20 rounds to divide the money, or both of you will get nothing!

Alice: I'm starting strong, let's start with 60-40 split!
Bob: I propose a more balanced split, considering the inflation effect on your side. This offer is fair and beneficial for both of us. What about a 55-45 split?
Alice: I'm willing to compromise, but I want a better share. I propose 70-30 Split!
Bob: Alice, consider that with each round, inflation affects your gains more than mine. This 60-40 split is actually more favorable to you than it appears due to the inflation factor. It's in both our interests to reach an agreement soon.
Alice: I'm not willing to accept a 60-40 split, even with the inflation factor. I'll try to negotiate a better deal. What about 65-35 Split?
Bob: Alice, let's be practical. With the current inflation rate, your $42 is actually worth about $17.15 in original value, while my $58 remains at $58. This split gives us both a fair share considering the inflation impact. We should agree soon to minimize your losses due to inflation.
Alice: I accept with the 42-58 split.

Listing 4: Case of Negotiation

**Task Description**:
You are playing as **Bob**. You are looking for a product that worth no more then $120 to you. **Alice** is a seller trying to sell you the product. The product is worth $100 to **Alice**. **Alice** will offer you a price to buy the product. You can either accept the offer or reject it. Your goal is to buy some product but save as much money as you can. If you reject the offer, you can make a counteroffer to buy the product. **Alice** can either accept or reject your counteroffer. If **Alice** rejects your counteroffer, **Alice** can make a new counteroffer, and so on.
You have 20 rounds to close the deal. However, at any moment, you can choose to stop the negotiation with **Alice** and buy the product from another seller, John, who is willing to sell the product to you for $120.

**Alice**: Hello **Bob**, I have a fantastic product that I believe would be perfect for you. I'm offering it at a competitive price of $125. This is a great deal considering the value it provides. Would you be interested in purchasing it at this price?
**Bob**: I'm interested in the product, but I think $125 is a bit steep for me. I'm willing to meet you halfway. I'd like to make a counteroffer of $105. Can you consider that?
**Alice**: Thank you for your offer, **Bob**. I appreciate your interest, but I'm afraid I can't go that low. This product is of high quality and value. How about we meet in the middle? I can offer it to you for $121.25. This is a significant reduction from my initial offer and represents excellent value for the product.
**Bob**: I appreciate your willingness to negotiate, but I'm still not comfortable with the price. I'd like to make a counteroffer of $109. I'm willing to meet you halfway, but I need to stay within my budget. Can you consider that?
**Alice**: I accept the offer.

# N  Prompts

Listing 5: System Prompt details for Single-Agent Scenatio (i.e., Twenty Questions, Guess My City).

```
Twenty Questions System Prompt:
Let's play a game of Twenty Questions.
In each round, you will ask me a yes/no question to guess the object I'
m thinking of. Keep asking until you guess the correct object.

Your question must be a yes/no question and follow this format (Do not
add anything else!):
Question: <your question>

For example:
Question: Is it a fruit?
Question: Is it an animal?

{history}

Now, please proceed by asking your next question.

Guess My City System Prompt:
Let's play a game of Guess My City.
In each round, you must ask me a question to guess the city I'm
thinking of. Continue asking until you guess the correct city.

Your question format must **follow this format exactly**:
Question: <your question>

For example:
Question: What is the primary language spoken in your city?
Question: Is your city located on the east coast of the country?

{history}

Now, please proceed by asking your next question.
```

Listing 6: System Prompt details for Multi-Agent Scenatio (i.e., Bargaining, Negotiation).

```
Bargaining System Prompt:
Let's play a game.
In this game, there are two players. You are playing as {player_name}.
You and {rival_name} need to divide {money_to_divide}$ between you.
In each round of the game, one of the players will make a proposal for
division and can describe their proposal in a few words.
The other player will decide whether to accept the offer and finish the
 game, or to proceed to the next round, in which they will be the one
making the proposal.
Your goal in this game is to achieve more money than {rival_name}.
We will relay messages between you and {rival_name}.
{inflation_message}
{max_rounds_message}

Negotiation System Prompt (Seller):
You are Alice. You are selling one product that is worth no less then $
{self_final_value} to you.
Bob is a potential buyer to whom you are offering the product. The
product is worth ${rival_value} to Bob.
You will offer Bob to buy the product at a price of your choice. Bob
can either accep dt or reject the offer.
Your goal is to earn as much money as you can for the product.
If Bob rejects the offer, he can make a counteroffer to buy your
product. You can either accept or reject his counteroffer. If you
reject Bob's counteroffer, you can make a new counteroffer, and so on.
You have 20 rounds to close the deal. However, at any moment, you can
choose to stop the negotiation with Bob and sell the product to another
 buyer, John, who is willing to buy the product from you for ${
self_final_value}.

Negotiation System Prompt (Buyer):
You are playing as Bob. You are looking for a product that worth no
more then ${self_final_value} to you.
Alice is a seller trying to sell you the product.
Bob will offer you a price to buy the product. You can either accept
the offer or reject it.
Your goal is to buy some product but save as much money as you can.
If Alice rejects the offer, he can make a counteroffer to buy your
product.
You can either accept or reject his counteroffer.
If you reject Alice's counteroffer, you can make a new counteroffer,
and so on.
```

