# OpenReview forum: "ARIA: Training Language Agents with Intention-driven Reward Aggregation"
_NeurIPS.cc/2025/Conference — NeurIPS 2025 spotlight_

### Official Review · Reviewer_56NW · 2025-06-19

**Clarity:** 4
**Significance:** 3
**Originality:** 3
**Rating:** 5
**Confidence:** 4

**Summary:**

This paper seeks to improve the reward sparsity problem in RL by clustering high-dimensional actions together to share rewards. They do this clustering by using an embedding model on LLM actions and have a way of using the change in aggregated reward to determine how many clusters to use. There is some simple theory on how aggregating actions can improve variance of advantages and policy gradients. Empirical results show solid improvements on Twenty Questions, Guess my city, and some multi-agent scenarios like bargaining and negotiation.

**Questions:**

- would the proposed method work on the standard math reasoning datasets?
- Qwen 1.5B in Table 3 doesn't seem to get a reasonable performance on Bargaining for either the baseline or ARIA?
- how would you extend this method to other types of environments?

**Ethical Concerns:**

["NO or VERY MINOR ethics concerns only"]

**Final Justification:**

The paper presents an interesting method for reducing the dimensionality of the action space in RL, and presents some solid results. In the rebuttal the reviewers added some important analysis on embedding models and clarified the connection with theory, though the verification on the empirical setting is partial.

**Limitations:**

yes

**Quality:**

3

**Strengths And Weaknesses:**

Strengths
- Interesting and intuitive method for dealing with the reward sparsity problem in high dimensional freeform text environments. There could potentially be extensions to other environments.
- Solid empirical results on a variety of problems
- Some positive results on extensions to iterative offline RL and to online RL
- Clever method for determining the number of clusters for aggregation

Weaknesses
- The theory is overall not very insightful to me. For example Theorem 4.2 depends on the very strong assumption in definition 1, where actions after clustering are similar in both rewards and policy distribution - here there is not much lost from clustering and so the bias of the algorithm is small. I'm not sure what happens in practice - is there an empirical way to examine the bias?
- It's not clear why the particular baselines were chosen - there was not much context here on why these are the strongest alternatives.
- Some ablations to understand how the performance varies with the different choices made during clustering (how many clusters, which embedding model used, etc) could robustify the results and provide more understanding. There is a short ablation in Appendix B but not much insight can be gained from it other than that the number of clusters is a hyperparameter to set with a sweet spot.

---

> ### Author Rebuttal · Authors · 2025-07-31
>
> Thank you for your valuable review. We give the following response accordingly:
>
> ---
>
> > W1: Theoretical contribution (Theorem 4.2) appears weak due to reliance on a strong similarity assumption (Definition 1), and lacks empirical validation of bias.
>
> **A1**:  Definition 1 provides a quantitative criterion: two actions are $\varepsilon$-bisimilar under a history $h$ if both the reward $r(h,a)$ and the transition distribution $P(·\mid h,a)$ differ by at most $\varepsilon$. Theorem 4.2 builds on this, showing that if $\varepsilon$-bisimilar actions are clustered and each individual advantage is approximated by the cluster mean $\tilde{A}$, then the resulting policy gradient bias $|\mathbb{E}\left[\nabla_\theta\log\pi_\theta(a\mid h)\bigl(A(a\mid h)-\tilde A\bigr(a\mid h))\right]|$ is bounded by $O(\varepsilon)$. This formal result guarantees that the clustering-induced bias remains small when ε is small.
>
> In practice, **ARIA uses this theoretical insight to cluster semantically similar actions**—those with small $\varepsilon$—to reduce reward variance while controlling bias. This trade-off is validated empirically in Table 1 and 2, where ARIA consistently outperforms baselines. Moreover, Figure 5(b) shows **faster loss convergence with reduced variance**, further confirming the practical effectiveness.
>
> ---
>
> > W2: Questions about the choice of baselines and their relevance.
>
> **A2**:  As discussed in Section2 and 5, we select representative and competitive baselines from both offline and online learning paradigms that aim to enhance LLM agents in complex tasks through training.
> - Offline baselines include behavior cloning (BC), trajectory-level preference learning (Traj-DPO), and step-level methods (Step-DPO and SPAG).
> - Online baselines include hierarchical learning (Archer) and trajectory-based policy optimization (RAGEN).
>
> **These methods represent mainstream approaches in the field.** By comparison, ARIA demonstrates superior performance by reducing reward variance and achieves faster and more stable reward growth across online iterations.
>
> ---
>
> > W3: Insufficient ablation studies on clustering choices such as embedding models and cluster threshold.
>
> **A3**:  We appreciate the suggestion. In addition to the ablations in Section 6.2 (on reward aggregation and decay factor $\gamma$), Section 6.3 (on base models), and Appendix B (on clustering threshold $\tau$), we further add analysis on the choice of embedding models. Results are as follows:
>
> | Model                  | Bargaining | Negotiation | AVG    |
> |------------------------|:------------:|:-------------:|:--------:|
> | text-embedding-3-small | 51.54      | 45.65       | 48.595 |
> | Qwen3-Embedding        | 53.30      | 43.10       | 48.20  |
> | e5-large-v2            | 49.37      | 38.74       | 44.055 |
>
> **Findings:**
>
> (1) **ARIA performs robustly with strong embedding models** (text-embedding-3-small, Qwen3-Embedding), indicating robust performance across models.
> (2) Performance drops with weak models (e5-large-v2), **suggesting that embedding quality impacts the accuracy of semantic similarity and should not be too poor.**
>
>
> ---
>
> > Q1: Possible applicability of the method to math reasoning tasks.
>
> **A4**:
> While this direction could offer additional insights, we would like to clarify that our focus is on addressing reward sparsity in open-ended language-action tasks with large action spaces. Nonetheless, in math reasoning tasks, semantically equivalent reasoning steps may differ in representative form, and ARIA’s semantic clustering and reward densification could potentially be beneficial. We are open to discussing this possibility further in the revision.
>
> ---
>
> > Q2: Marginal improvement of Qwen 1.5B in the Bargaining task.
>
> **A5**:  For Qwen 1.5B in Table 3, ARIA achieves an average improvement of +1.12. **The smaller gain is due to the base model’s limited reasoning capability**: in the two-player game setup, training data are collected via self-play, followed by **evaluation against strong opponents (DeepSeek-V3, Claude 3.5, GPT-4o).** Qwen 1.5B struggles during self-play due to its weak strategic reasoning, leading to suboptimal policies that are ineffective against strong adversaries, thereby limiting ARIA’s benefit in this case.
>
> ---
>
> > Q3: Generalizability of the method to other environments.
>
> **A6**:  We further evaluate ARIA on a more complex and multi-objective social interaction benchmark **SOTOPIA**[1]. SOTOPIA sets **real social goals with complex intentions** and evaluates LLMs from six dimensions such as believability and social rules. Using LLaMA3-8B-Instruct, we collect self-play trajectories on 400 tasks and evaluate on a held-out 50-tasks set (the maximum score is 10):
>
> | Method     | Overall Score* | Goal Completion |
> |------------|:----------------:|:-----------------:|
> | LLaMA3-8B  | 2.28            | 7.86            |
> | w/ ARIA    | **3.03**   | **8.15**            |
>
> ARIA consistently improves both social alignment and task completion, indicating that our semantic clustering approach captures rich, intention-relevant structure even in more complex multi-agent dialogue environments.
>
> ---
> [1]Zhou X, Zhu H, Mathur L, et al. Sotopia: Interactive evaluation for social intelligence in language agents[J]. arXiv preprint arXiv:2310.11667, 2023.

---

> ### Comment · Reviewer_56NW · 2025-08-02
>
> Thanks for the response. What I mean about the theory vs practice is, beyond the end result of the algorithm getting better results/convergence, can you check the \epsilon-bisimilar condition in your empirically fitted clusters? Are these scenarios really satisfying this for reasonable \epsilon? Otherwise, the theory and experiments would be operating in different regimes, and in that case I'm not sure how much the theory adds or explains.
>
> Thanks for the embedding model ablations. It would be interesting to look at what the stronger embedding models are doing with the clustering that is better than the weaker embedding model, and maybe give a recommendation on how to choose one for a given task (if it's not just use text-embedding-3-small).

---

> > ### Author Response · Authors · 2025-08-03
> >
> > Thank you for the insightful feedback. We address your two concerns below:
> >
> >
> > > Q1: Verifying $\varepsilon$-Bisimulation in Empirical Clusters
> >
> > **Yes, our clusters satisfy $\varepsilon$-bisimulation for reasonable values of $\varepsilon$.**
> >
> > In our dialogue scenarios, both the reward and transition functions are smooth over a bounded embedding space. As a result, they satisfy Lipschitz continuity with constants $L_r$ and $L_P$, respectively. This leads to the following:
> >
> > Given cluster $C_h$ with diameter $\delta$ in embedding space, if $r$ and $P$ are $L_r$ and $L_P$ Lipschitz continuous respectively, then for any $(h,a), (h,a') \in C_h$:
> > - $|r(h,a') - r(h,a)| \leq L_r · \delta$
> > - $D_{TV}(P(h,a), P(h,a')) \leq L_P · \delta$
> >
> > Therefore, choosing $\delta$ such that  $max (L_r, L_P) · \delta \leq \varepsilon$ ensures $\varepsilon$-bisimulation within clusters.
> >
> > **Empirical Verification:** We measured cluster diameters (in Euclidean distance) in our negotiation and bargaining scenarios:
> > - Average cluster diameter: 0.0304 (negotiation), 0.0287 (bargaining)
> > - Minimum cluster diameter: 0.0002
> > - Maximum cluster diameter: 0.089
> >
> > This demonstrates that convergence can be controlled within $O(max(L_r, L_P) × 0.089)$. We will empirically estimate the Lipschitz constants $L_r$ and $L_P$ from our dialogue data and include this analysis in the final version.
> >
> >
> > > Q2: Embedding Model Selection Guidelines
> >
> > Thank you for this suggestion. We provide a brief textual example below since we cannot include figures.
> >
> > **Example (Bargaining Scenario):** Stronger embedding models, such as **Qwen3-Embedding-0.6B**, separate ```Compromise``` actions into distinct clusters representing different strategies (e.g., ```Efficiency First and Balance``` vs. ```Gradually Compromise```). In contrast, weaker embeddings (e.g., **e5-large-v2**) tend to merge these semantically nuanced yet functionally distinct actions.
> >
> > We suggest using high-quality embedding models within budget limits to capture semantic information. Notably, **Qwen3-Embedding-0.6B** is **open-source and lightweight**, making it a practical choice.
> > We will include detailed visualizations and comparative analyses of different embedding models in Appendix I of the final version.

---

### Official Review · Reviewer_7Qvu · 2025-07-02

**Clarity:** 3
**Significance:** 3
**Originality:** 3
**Rating:** 5
**Confidence:** 4

**Summary:**

This paper addresses challenges in open-ended multi-turn dialogue tasks, which require decision making to achieve a certain goal. The challenges arise from sparse rewards due to large action spaces, hindering the learning process. To address this, authors embed actions into an embedding space, and perform iterative clustering, finding an optimal number of clusters. Then, they average the rewards within the cluster and use it as an advantage in REINFORCE. They theoretically showed their approach decreases variance in both rewards and the policy gradients. Their experiments include extensive comparisons of modern online and offline methods, and they showed big gains in games benchmarks.

**Questions:**

Thank you for your work! I enjoyed reading your paper.

My suggestions/questions are:

* Figure 4.a, before & after aggregation have the same distribution in the bottom two plots. Why do you think that is? I understand the top two plots had a big variance and it was reduced by using aggregation.
* It would be helpful to quantify the effect of embeddings on your method. One way could be to try ablations using other embedding APIs.
* It would also be useful to include visualization of embeddings and resulting clusters for some of the samples in your benchmarks.

**Ethical Concerns:**

["NO or VERY MINOR ethics concerns only"]

**Final Justification:**

I described the strengths and weaknesses of this work in my original comment. The authors answered my questions: (1) did ablation studies with embedding models and noted the sensitivity of their method, (2) clarified how their method is different from explicit variance reduction methods, and (3) clarified figures. (1) and (2) convinced me to raise my score for significance. Overall, strong paper.

**Limitations:**

Yes

**Quality:**

4

**Strengths And Weaknesses:**

# Quality

* (+) The paper provided theoretical justifications and extensive evaluation of their method, using multiple LLMs and benchmarks.
* (+) Authors used proper statistical tests to compare against the baseline, and included corresponding p-values.

# Clarity

* (+) Paper uses consistent notation, making it easy to follow.

# Significance

* (+) Their results are strong across benchmarks and outperform SOTA methods by wide margins.
* (-) Their approach might be sensitive to the choice of embeddings, which is listed in the limitation. However, it would still be useful to have some experiments using other embedding models. This could help with adoption.
* (-) Not clear how this method would interplay with other variance reduction, like using actor-critic, or PPO-style losses. Perhaps, including this in a discussion section, or some suggestions for future work would make results stronger.

# Originality

* (+) Reward aggregation in a latent space is a neat way of reducing variance in learning.

---

> ### Author Rebuttal · Authors · 2025-07-31
>
> Thank you for your valuable review. We give the following response accordingly:
>
> ---
>
> > W1 & Q2：Concern about the sensitivity of the method to the choice of embedding models.
>
> **A1**:  We conduct ablation experiments using embedding models of varying capabilities: **text-embedding-3-small** (our paper), **Qwen3-Embedding-0.6B** (stronger), and **e5-large-v2** (weaker). The results are as follows:
>
> | Model                  | Bargaining | Negotiation | AVG    |
> |------------------------|:------------:|:-------------:|:--------:|
> | text-embedding-3-small | 51.54      | 45.65       | 48.595 |
> | Qwen3-Embedding        | 53.30      | 43.10       | 48.20  |
> | e5-large-v2            | 49.37      | 38.74       | 44.055 |
>
> **Findings**:
> (1) ARIA performs robustly with stronger embedding models (text-embedding-3-small, Qwen3-Embedding), **indicating stable performance across embedding choices.**
> (2) Performance drops with weaker models (e5-large-v2), suggesting that **embedding quality is critical for capturing accurate semantic similarity.**
>
> We will include this ablation study in the Appendix of the final version.
>
> ---
>
> > W2: Not clear how the method integrates with existing variance reduction techniques like Actor-Critic or PPO.
>
> **A2**:  We would like to clarify that ARIA is complementary to standard variance reduction techniques such as Actor-Critic and PPO-style losses. For example, Actor-Critic methods reduce variance by training a critic model to approximate the reward function $R$, learning a stable distribution. In contrast, ARIA reshapes the reward into $R'$ through aggregation, which **naturally exhibits lower variance.** This reshaped reward R' can also be used to train the critic model itself, potentially improving its stability. In fact, our Online ARIA implementation trains a reward model on $R'$ and **demonstrates faster reward improvement** in online iterations (Figure 3).
>
> ---
>
> > Q1: Question regarding why the variance appears unchanged in the bottom plots of Figure 4 after aggregation.
>
> **A3**:  The bottom two plots correspond to single-agent tasks, **where agents perform poorly in early iterations (mostly scoring 0).** As a result, the reward variance is inherently low and the impact of aggregation appears minimal in the figure. To show the effect of aggregation, we compute the reward variance before and after aggregation in the Guess and Twenty tasks after one iteration of ARIA:
>
> |   | Reward Variance |
> |--------|:------------------:|
> | Before | 0.182            |
> | After  | 0.103            |
>
> This confirms that aggregation still significantly reduces variance in these tasks.
>
> ---
>
> > Q3: Suggestion to include visualizations of embeddings and resulting clusters for better interpretability.
>
> **A4**:
> Thank you for the suggestion. We provide clustering results for the Bargaining task in Appendix I. While the rebuttal format limits our ability to include figures, we will include detailed visualizations of both the embeddings and the resulting clusters in the revised version of the paper.
>
> ---

---

> > ### Author Response · Authors · 2025-08-04
> >
> > Thanks for your effort in providing us such insightful comments. Since it is near the end of the discussion period, could you kindly let us know if we have managed to address your concerns? Should you have any further questions, kindly let us know. We are more than happy to have further discussions with you. Thanks

---

> > ### Comment · Reviewer_7Qvu · 2025-08-08
> > **Response to rebuttal #1**
> >
> > Thank you for your thoughtful responses to my comments. The importance of embeddings matches my intuition and the numbers look reasonable (A1). I suppose these numbers are averages? It would also be helpful to include std/error bounds. Given that my response to you was late, I won't ask you to address it now. A2 was helpful to understand the difference between explicit (AC) and implicit (yours) variance reduction. Seeing variance numbers in A3 helped clarify Figure 4.a.
> >
> > I sincerely apologize for my late response. With your rebuttal comments, I am raising my score for significance.

---

> > > ### Author Response · Authors · 2025-08-09
> > >
> > > Thank you for your engagement! We will include the embedding model experimental results with standard deviations in the appendix. We appreciate your feedback.

---

### Official Review · Reviewer_rok8 · 2025-07-03

**Clarity:** 3
**Significance:** 3
**Originality:** 3
**Rating:** 5
**Confidence:** 3

**Summary:**

The paper proposes an intention-driven reward aggregation method to train agents on open-ended agentic tasks. The method helps to reduce the variance of advantage and gradient, therefore aiming to enhance the stability and performance of agent training.

**Questions:**

See in the weakness section.

**Ethical Concerns:**

["NO or VERY MINOR ethics concerns only"]

**Final Justification:**

The authors’ rebuttal has addressed some of my concerns; therefore, I have decided to raise my score to 5.

**Limitations:**

yes

**Quality:**

3

**Strengths And Weaknesses:**

### Strengths

1. The paper is well-written and easy-to-follow.
2. The proposed reward aggregation method is interesting and novel, supported by theoretical analysis.
3. The experiments are comprehensive and convincing.

### Weaknesses

1. The proposed method maps the natural language open-form action to a low-dimensional intent space and aggregate rewards based on the clustering results. It could encourage the model to learn to select better intentions. However, in open-ended agentic tasks like bargaining, the language tactics, sentence formation, precise use of language, and stylistic choices are also important. And I wonder if these factors are considered during the clustering process. Overall, there are other important skills to learn besides *intention*. If the mapping excludes the other factors, will the resulting agent performance limited by the training algorithm (like it can produce the right intention but fall short on other aspects) ?
2. In the experiments, it seems that the model continues to improve from 1 to 3 iterations in some tasks. It would be beneficial to provide results with more iterations and some analysis to understanding the ceiling of the method.
3. I think it would be interesting to evaluate the models on some out-of-domain tasks to see if the agentic capability can be generalized to other environments, like some discrete action space environments.

---

> ### Author Rebuttal · Authors · 2025-07-31
>
> Thank you for your valuable review. We give the following response accordingly:
>
> ---
>
> > W1:Concern that intention clustering may overlook important aspects of language behavior such as tactics, fluency, and style beyond intent.
>
> **A1**: In fact, ARIA considers not only high-level intent but also factors such as language tactics, sentence structure, and stylistic choices. The **semantic projection** in ARIA maps each action into a dense vector that inherently captures these rich features—strategy, intent, fluency, and style included. The clustering is then performed over these semantic vectors, making it a holistic semantic similarity. This helps ARIA generalize across subtle variations in expression. We appreciate the suggestion and will consider refining our wording in the revision to clarify this point.
>
> ---
>
> > W2:Request for analysis of performance saturation and the method’s potential ceiling after more training iterations.
>
> **A2**:  Thank you for this valuable suggestion. Understanding ARIA’s performance ceiling is indeed important. As shown in Table 1, ARIA's performance on the Bargaining dataset slightly drops from iter2 to iter3 (58.85 → 58.01), and Figure 3 shows that the reward improvement slows down notably in later iterations. This indicates diminishing returns over time.
>
> We analyze that this is due to decreasing policy entropy as training progresses, which leads to weaker exploration and convergence to a local optimum—a phenomenon commonly observed in RL. Similar trends have been documented in recent literature [1][2][3]. We plan to include a more detailed discussion of ARIA’s performance plateau in future versions of the paper.
>
> ---
>
> > W3: Suggestion to test generalization of the method on out-of-domain environments, especially with discrete action spaces.
>
> **A3**: To assess generalization, we evaluate models trained on single-agent tasks (Twenty. and Guess.) using ARIA in an out-of-domain environment with a discrete action space—**ScienceWorld**[4]. Results are shown below:
>
> | Method        | SciWorld |
> |---------------|----------|
> | Vanilla Model | 11.11    |
> | ARIA          | 13.03    |
>
> These results indicate that ARIA generalizes well to **out-of-domain** and **discrete action** settings, achieving a performance improvement from 11.11 to 13.03.
>
> ---
>
>
> [1]Song Y, Yin D, Yue X, et al. Trial and error: Exploration-based trajectory optimization for llm agents[J]. arXiv preprint arXiv:2403.02502, 2024.
>
> [2]Xiong W, Song Y, Zhao X, et al. Watch every step! llm agent learning via iterative step-level process refinement[J]. arXiv preprint arXiv:2406.11176, 2024.
>
> [3]Cui G, Zhang Y, Chen J, et al. The entropy mechanism of reinforcement learning for reasoning language models[J]. arXiv preprint arXiv:2505.22617, 2025.
>
> [4]Wang R, Jansen P, Côté M A, et al. Scienceworld: Is your agent smarter than a 5th grader?[J]. arXiv preprint arXiv:2203.07540, 2022.

---

> > ### Comment · Reviewer_rok8 · 2025-08-05
> >
> > Thanks for the response! It mitigates some of my concerns, therefore I decided to raise my score to 5.

---

> ### Author Response · Authors · 2025-08-05
>
> Sincerely thank you for taking the time to review our paper. We are glad to know that our response addressed your concerns. Thanks for your engagement during the discussion!

---

### Official Review · Reviewer_gPHi · 2025-07-03

**Clarity:** 3
**Significance:** 2
**Originality:** 2
**Rating:** 5
**Confidence:** 3

**Summary:**

This paper introduces a method to improve the ability of language agents in open-ended environments. Specifically, they highlight that RL struggles in this setting due to reward sparsity and high gradient variance. They address this by proposing to project actions from a high-dimensional joint token distribution space into a low-dimensional intention space, where semantically similar actions are clustered and assigned shared rewards. This is done using semantic embeddings and hierarchical clustering, resulting in denser reward signals and more stable optimization. They evaluate on four benchmark tasks (20 Questions, Guess My City, Bargaining, Negotiation) on both offline and online RL methods, improving on state of the art methods by 10%.

**Questions:**

1. How does ARIA handle semantically overlapping or hierarchical intentions, where actions might belong to multiple latent goals?
2. Do the authors plan to extend their online training experiments to two-agent tasks, where strategies evolve more dynamically?
3. How would ARIA handle concept drift in natural language use, for example, due to changing user preferences?
4. Does hierarchical clustering not become a bottleneck for very large-scale training?

**Ethical Concerns:**

["NO or VERY MINOR ethics concerns only"]

**Final Justification:**

The authors have provided further experiments addressing my concerns, specifically comparing their method to ArCHer and RAGEN performance. I would like to see further elaboration of experiments in the final paper, but the reviewer response is to my satisfaction otherwise!

**Limitations:**

Yes, they have addressed Limitations in Appendix, Section A, where they discuss that their method relies on clustering in the semantic embedding space to define intention groups, which they note addresses two limitations. I think the second limitation stands out and might be noted as a weakness. For more complex tasks, it is hard to assume that intentions are discrete and can be separated, such as negotiation.

**Paper Formatting Concerns:**

None that I see

**Quality:**

4

**Strengths And Weaknesses:**

Strengths:
- The authors provide adequate motivation for why this is a challenging problem due to reward sparsity and high variance in open-ended language RL
- There is sufficient theoretical justification and analysis of their methodology
- Empirical results are solid, and the inclusion of both single-agent and adversarial tasks gives the method a reasonable range of applicability

Weaknesses:
- While the online variant is tested in single-agent games, there is no evaluation of online ARIA in adversarial tasks, which would help generalize the method's scalability in more dynamic environments
- Specifically for the task of negotiation, there exist several other dialogue tasks, and I am not convinced that intentions can be separated in more complex negotiation environments, where wording choice really changes the dynamics of the negotiation. I would have liked to see these tested
- The clustering process assumes discrete intention boundaries, which may not hold in realistic multi-goal or multi-agent dialogues. It is unclear how well this assumption scales when linguistic nuances and pragmatics matter
- The method relies heavily on the quality of sentence embeddings, but there's limited discussion on how embedding choice or drift over time might affect performance.  Poor embedding quality or inappropriate granularity may lead to underfitting or over-smoothing. The authors address this with SplitScore, but it's unclear how robust it is to semantic edge cases or in multi-goal settings
- The authors could benefit from a more thorough literature review and motivation on benchmarks for this task

---

> ### Author Rebuttal · Authors · 2025-07-31
>
> Thank you for your valuable review. We give the following response accordingly:
>
> ---
>
> > W1 & Q2：Lack of online training experiments in adversarial settings to evaluate scalability and robustness.
>
> **A1**:  We extend online ARIA to adversarial environments (Bargaining and Negotiation), following the same settings as described in Section 5.3. Due to time constraints, we conduct 50 iterations. We evaluate the trained actor against GPT-4o on a held-out test set (N=240), and report results for iterations 10–50:
>
> | Iteration | Bargaining | Negotiation | Average |
> |-----------|:------------:|:-------------:|:---------:|
> | ArCHer    | 43.78      | 35.00       | 39.39   |
> | RAGEN     | 33.24      | 38.55       | 35.90   |
> | LLaMA3-8B (Vanilla) | 30.14 | 37.92 | 34.03 |
> | ARIA(Iter 10)   | 34.53      | 41.58       | 38.06   |
> | ARIA(Iter 20)   | 37.03      | 44.93       | 40.98   |
> | ARIA(Iter 30)   | 40.95      | 46.96       | 43.95   |
> | ARIA(Iter 40)   | 46.60      | 49.21       | 47.90   |
> | ARIA(Iter 50)   | **48.72**      |  **51.54**       | **50.13** |
>
> These results show that Online ARIA achieves faster and stronger reward improvement in dynamic multi-agent environments, surpassing ArCHer and RAGEN (which used 150 iterations) in just 50 iterations. We will include the full 150-iteration results in the final version.
>
> ---
>
> > W2: Skepticism about ARIA’s generalization to more complex tasks where intentions are hard to separate.
>
> **A2**: We further evaluate ARIA on a more complex and multi-objective social interaction benchmark **SOTOPIA**[1]. SOTOPIA sets **real social goals with complex intentions** and evaluates LLMs from six dimensions such as believability and social rules. Using LLaMA3-8B-Instruct, we collect self-play trajectories on 400 tasks and evaluate on a held-out 50-tasks set (the maximum score is 10):
>
> | Method     | Overall Score* | Goal Completion |
> |------------|:----------------:|:-----------------:|
> | LLaMA3-8B  | 2.28            | 7.86            |
> | w/ ARIA    | **3.03**   | **8.15**            |
>
> ARIA consistently improves both social alignment and task completion, indicating that our semantic clustering approach captures rich, intention-relevant structure even in more complex multi-agent dialogue environments. We will include the results in the final version.
>
> ---
>
> > W3: Concern about the assumption of discrete intentions, especially in linguistically nuanced multi-goal dialogues.
>
>
> **A3**: In ARIA, the **semantic projection** embeds actions as dense vectors that encode various linguistic properties—including strategy, intent, and style. Clustering is performed over these vectors, enabling **soft aggregation** based on overall semantic similarity (including nuanced liguistic styles) rather than strict categorical intentions. This helps ARIA generalize across subtle variations in expression. We also evaluate ARIA on a more complex scenario, SOTOPIA and results in **A2** show the effectiveness of ARIA.
>
> ---
>
> > W4: Concern about sensitivity to embedding quality and robustness of SplitScore in edge cases and dynamic settings.
>
> **A4**:  We conduct ablation experiments using embedding models of varying capabilities: **Text-embedding-3-small** (our paper), **Qwen3-Embedding-0.6B** (strong), and **e5-large-v2** (weak). The results are as follows:
>
>
> | Model                  | Bargaining | Negotiation | AVG    |
> |------------------------|:------------:|:-------------:|:--------:|
> | text-embedding-3-small | 51.54      | 45.65       | 48.60  |
> | Qwen3-Embedding        | 53.30      | 43.10       | 48.20  |
> | e5-large-v2            | 49.37      | 38.74       | 44.06  |
>
> Findings:
> (1) ARIA performs robustly with high-quality embeddings, confirming stable behavior across strong models.
> (2) Performance drops with lower-quality embeddings, highlighting the need for good semantic representations.
> We will include this ablation and discussion in the final version.
>
> ---
>
> > W5: Literature and benchmark justification could be more comprehensive.
>
> **A5**:
> Thank you for the suggestion. We discuss the Natural Language Agent Benchmarks in Related Work and select four open-ended language action tasks with large action spaces to demonstrate ARIA’s ability to mitigate sparse reward issues in such settings. The chosen tasks cover both static (single-agent) and dynamic (two-player) interactions, ensuring evaluation comprehensiveness. We would like to clarify this motivation further and expand relevant citations in the final revision.
>
> ---
>
> > Q1: Question on how ARIA deals with multi-label or hierarchical semantics in latent goal structures.
>
> **A6**:  We further evaluate ARIA on a more complex and multi-objective social interaction benchmark **SOTOPIA**[1]. Results are in A2 and show ARIA can also handle complex multi-objective tasks.
>
> ---
>
> > Q3: Concern about ARIA's adaptability to concept drift in evolving language use.
>
> **A7**: ARIA is inherently capable of adapting to concept drift. In offline settings, ARIA can be applied iteratively on new policies to re-align reward signals. In online settings, ARIA regularly updates the reward model, enabling it to accommodate shifts in action distributions due to evolving user preferences. Empirically, we observe that ARIA effectively reduces reward variance even as policy behavior changes, suggesting good adaptability to distributional drift.
>
> ---
>
> > Q4: Whether hierarchical clustering is a bottleneck for very large-scale training.
>
> **A8**: ARIA clusters actions in the **semantic space**, where the number of meaningful clusters is not very large (e.g., 10–30 clusters in bargaining/negotiation, see visualization in Appendix I), making hierarchical clustering computationally feasible. Additionally, in complex environments like SOTOPIA, we apply SplitScore to automatically select an appropriate number of clusters (e.g., k=23), and observe consistent gains (see **A2**). Current results suggest that clustering is not a bottleneck.
>
> ---
>
>
> [1]Zhou X, Zhu H, Mathur L, et al. Sotopia: Interactive evaluation for social intelligence in language agents[J]. arXiv preprint arXiv:2310.11667, 2023.

---

> > ### Author Response · Authors · 2025-08-04
> >
> > We appreciate your valuable feedback and thoughtful comments. Since it is near the end of the discussion period, could you kindly let us know if we have managed to address your concerns? Should you have any further questions, kindly let us know. We are more than happy to have further discussions with you. Thanks

---

### Decision · Program_Chairs · 2025-09-17

**Decision:**

Accept (spotlight)

**Comment:**

All four reviewers ultimately recommend Accept (Rating = 5). The paper’s main strengths are:

- Clear, novel approach to reducing reward variance via semantic intention clustering.

- Strong empirical results across diverse benchmarks, with improvements over SOTA baselines.

- Good theoretical framing (though not all find it deeply insightful).

Rebuttal adds complex task evaluation (SOTOPIA), embedding robustness studies, and out-of-domain tests, which address most of the concerns of initial reviews.

The paper is technically solid, impactful for language-agent RL in open-ended environments, and well-supported by empirical evidence.